# MYC functions as a switch for natural killer cell-mediated immune surveillance of lymphoid malignancies

Srividya Swaminathan[1,4,5], Aida S. Hansen [1,5], Line D. Heftdal [1], Renumathy Dhanasekaran[1,2], Anja Deutzmann[1], Wadie D. M. Fernandez[1], Daniel F. Liefwalker [1], Crista Horton[1], Adriane Mosley[1], Mariola Liebersbach[1], Holden T. Maecker[3] & Dean W. Felsher [1✉]

The *MYC* oncogene drives T- and B- lymphoid malignancies, including Burkitt's lymphoma (BL) and Acute Lymphoblastic Leukemia (ALL). Here, we demonstrate a systemic reduction in natural killer (NK) cell numbers in *SRα-tTA/Tet-O-MYC^{ON}* mice bearing MYC-driven T-lymphomas. Residual mNK cells in spleens of *MYC^{ON}* T-lymphoma-bearing mice exhibit perturbations in the terminal NK effector differentiation pathway. Lymphoma-intrinsic MYC arrests NK maturation by transcriptionally repressing *STAT1/2* and secretion of Type I Interferons (IFNs). Treating T-lymphoma-bearing mice with Type I IFN improves survival by rescuing NK cell maturation. Adoptive transfer of mature NK cells is sufficient to delay both T-lymphoma growth and recurrence post MYC inactivation. In MYC-driven BL patients, low expression of both STAT1 and STAT2 correlates significantly with the absence of activated NK cells and predicts unfavorable clinical outcomes. Our studies thus provide a rationale for developing NK cell-based therapies to effectively treat MYC-driven lymphomas in the future.

[1] Division of Oncology, Departments of Medicine and Pathology, Stanford University, Stanford, CA, USA. [2] Division of Gastroenterology and Hepatology, Stanford University, Stanford, CA, USA. [3] The Human Immune Monitoring Center (HIMC), Institute for Immunity, Transplantation and Infection, Stanford University School of Medicine, Stanford, CA, USA. [4] Present address: Department of Systems Biology, Beckman Research Institute of City of Hope, Duarte, CA, USA. [5] These authors contributed equally: Srividya Swaminathan, Aida S. Hansen. ✉email: dfelsher@stanford.edu

Oncogene addiction is the phenomenon where cancers become dependent on a single 'driver' oncogene for proliferation and survival[1,2]. Inhibition of the driver forms the basis of oncogene-targeted therapy[1–3]. Hence, a potential vulnerability of human malignancies lies in their addiction to oncogenes including *MYC*, *RAS* and *BCR-ABL1*[4,5]. Targeted therapy has been most successful against BCR-ABL1 (Philadelphia/Ph+)-induced leukemia[6,7]. To develop targeted therapies for MYC-driven cancers, it is vital to understand how MYC regulates cell-autonomous and non-autonomous processes, including host immunity. Oncogene addiction was assumed to be cell-autonomous and immune-independent[8]. Recent studies illustrate that MYC and other oncogenes alone or cooperatively regulate the tumor microenvironment and host immune responses in multiple tumor types[9–16].

We use tractable approaches for studying the role of the host immunity during MYC-driven tumorigenesis through tetracycline (tet)-system regulated transgenic mouse models of cancer. These models enable us to understand how MYC inactivation elicits tumor regression through cancer-intrinsic and extrinsic host immune-dependent mechanisms[1,10,15]. CD4+ T cells appear to be essential for sustained cancer regression post MYC inactivation[10]. However, changes in immune landscape during primary MYC-induced tumorigenesis in these models remain to be delineated.

We aimed to delineate global immunological changes resulting from primary overt MYC-driven lymphomagenesis to identify anti-tumor immune subsets that can be developed as immunotherapies against MYC-driven lymphomas. We hypothesized that MYC directly perturbs host immunity at sites of lymphomagenesis to evade immune surveillance. Hence, using mass cytometry (CyTOF), we examine specific changes in the host immune composition upon MYC activation as well as after subsequent MYC inactivation in situ in *SRα-tTA/tet-O-MYC* mice predisposed to developing MYC-driven T cell lymphoblastic lymphoma[1].

We observe that MYC suppresses maturation of natural killer (NK) cells in the lymphoma microenvironment. We find that NK cells have a key anti-tumorigenic role by delaying the growth and recurrence of MYC-driven T-lymphomas, and are hence suppressed by MYC during lymphomagenesis. We show a direct signaling mechanism by which lymphoma-intrinsic MYC suppresses NK cell-mediated immune surveillance. Our results provide a rationale for developing and combining NK cell-based therapies with MYC inhibitors to treat MYC-driven lymphomas.

## Results

### MYC-driven lymphomas exhibit disrupted splenic architectures.

We examined gross changes in spleen resulting from overt MYC-driven T-lymphomagenesis in *SRα-tTA/tet-O-MYC* mice[1]. Of note, SRα restricts the overexpression of the human MYC (*hMYC*) transgene to the T cell lineage in *SRα-tTA/tet-O-MYC* mice, giving rise to systemically disseminated T-lymphomas[1]. Lymphoma-bearing mice displayed splenic germinal center disruption, whereas MYC inactivation partially rescued the splenic architecture (Supplementary Fig. 1). Therefore, we speculated that MYC overexpression in T-lymphoma might remodel the splenic immune landscape.

### Oncogenic MYC perturbs frequencies of splenic immune subsets.

Inducible regulation of the *hMYC* transgene specifically in T-lymphoblasts enables us to elucidate how lymphoma-intrinsic MYC impacts normal immune cells during primary lymphomagenesis. Using CyTOF[17], we delineated the global immunological changes in lymphoid organs during primary MYC-induced lymphomagenesis in *SRα-tTA/tet-O-MYC* mice.

Splenic samples derived from overt lymphoma-bearing *SRα-tTA/tet-O-MYC* mice before (*MYC^ON*) and after (*MYC^OFF*, 96 h) MYC inactivation were examined for immune subsets; and compared to age- and strain-matched normal spleens (normal).

Consistent with the phenotype of T-lymphoma, the distribution of T-lymphocyte subsets was altered in the *MYC^ON* cohort with an increase in percentages of immature CD4+CD8+ double positive (DP) CD3+ T-lymphoblasts as compared to normal mice. MYC inactivation resulted in elimination of most DP T-lymphoblasts and restored distribution of splenic T-subsets to normal levels, demonstrating MYC-addiction, as previously described[1] (Fig. 1a–c, Supplementary Fig. 2a, b). T-lymphoblasts in *MYC^ON* mice are TCRαβ+, thus leading to a reduction in the percentages of TCRγδ+ T cells (Supplementary Fig. 2c, d).

We next examined other non-malignant immune compartments in *MYC^ON and MYC^OFF* mice, and compared these to normal mice. The percentages of NK (CD3−NKp46+), NKT (CD3+NKp46+) and B cells (CD19+) were significantly lowered in *MYC^ON* mice, and were restored close to normal levels in *MYC^OFF* mice (Fig. 1d, e, Supplementary Figs. 2e–g and 3a, b) The relative proportions of other immune compartments including dendritic cells (DCs) and neutrophils were unaltered by modulation of MYC (Supplementary Fig. 3c–f).

### Immune changes in T-lymphomas are independent of splenomegaly.

Lymphomagenesis is often associated with increased splenic cellularity. To rule out that the apparent reduction in NK and B cells was because they were being passively outnumbered by T-lymphoblasts, we measured the absolute cell numbers of immune subsets in splenic samples evaluated by CyTOF. We observed significant increases in the cellularity of lymphoma spleens (Fig. 1f). As expected, absolute counts of CD3+ pan T, TCRαβ+CD3+ T and immature DP CD3+ T-lymphoblasts were increased in *MYC^ON* mice when compared to normal and *MYC^OFF* mice (Fig. 1g, Supplementary Fig. 4a, b). Characteristic with lymphoma, multiplication of the TCRαβ+CD3+ T-lymphoblasts was accompanied by a significant reduction in CD3+TCRγδ+ T-lymphocyte numbers in *MYC^ON* mice (Supplementary Fig. 4c).

Oncogenic MYC significantly lowered numbers of CD3− NKp46+ NK and CD3+NKp46+ NKT cells, whereas MYC inactivation reversed this effect (Fig. 1h, Supplementary Fig. 4d). Despite reduction in B-lymphocyte frequency, numbers of B cells were unaltered in MYC-driven lymphomas as compared to normal and *MYC^OFF* mice (Fig. 1i). MYC-driven lymphomagenesis significantly increased numbers of DCs and neutrophils (Supplementary Fig. 4e, f), corroborating the previously well-characterized pro-tumorigenic functions of these subsets in lymphomas[18–20].

### NK cells are specifically suppressed in MYC-driven T-lymphomas.

Our goal was to identify anti-tumor immune subsets that can be developed as therapies against MYC-driven lymphomas[21,22]. As NK cells are attractive as a potential cell-based immunotherapy against MYC-driven lymphomas, we continued to focus specifically on how MYC alters the NK subset during lymphomagenesis. We confirmed our CyTOF results by measuring NK compositions before and after *MYC* inactivation by conventional flow cytometry (Supplementary Fig. 5). Next, using a computational method for estimating immune compositions from bulk gene expression data (CIBERSORT[23]), we calculated changes in NK composition in CyTOF-matched spleens

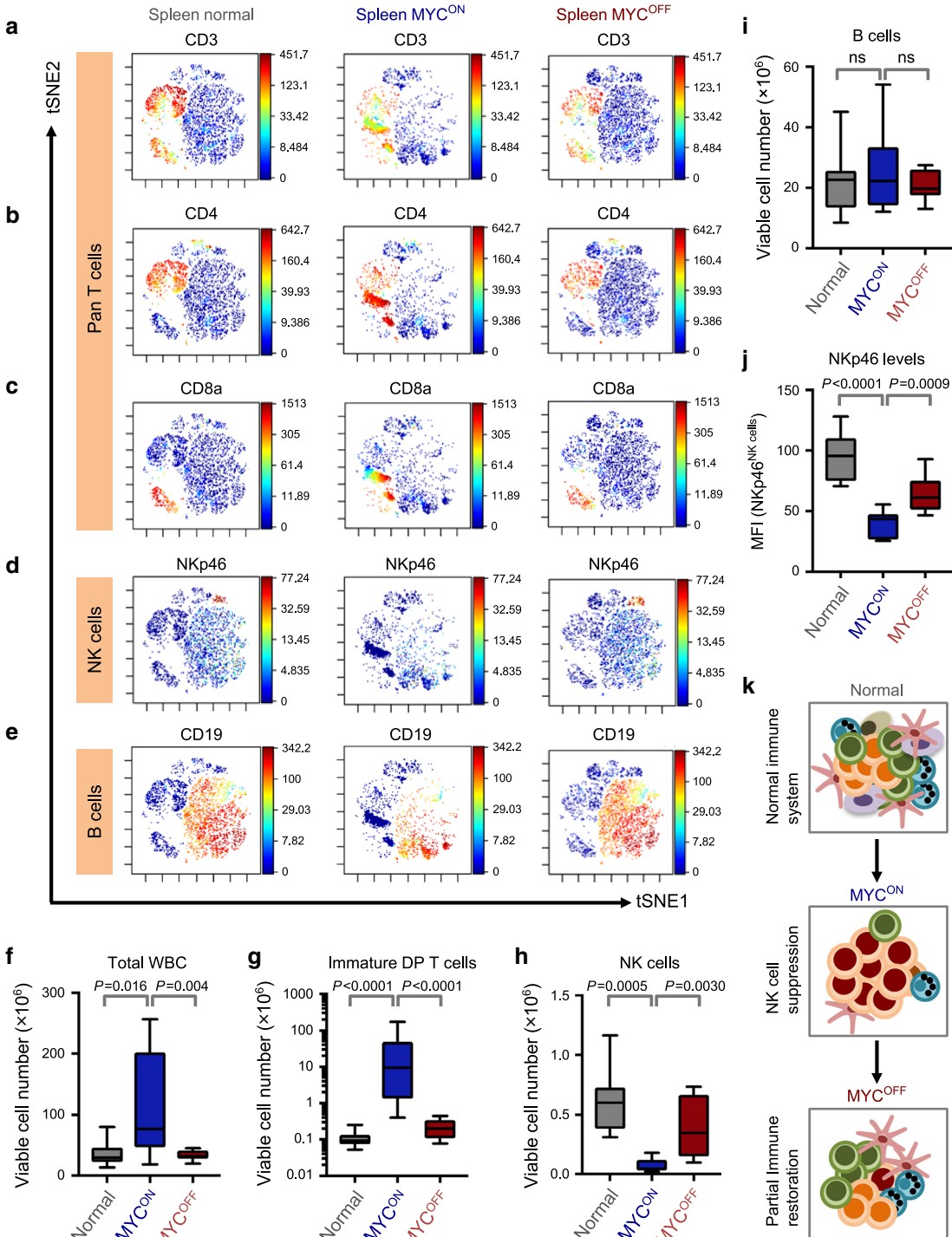

**Fig. 1 NK cells are reduced in numbers and maturation in MYC-driven T-lymphomas. a–e** viSNE of CyTOF data depicting splenic CD3+ T (**a**), splenic CD4+ T (**b**), CD8a+ T (**c**), NKp46+ NK (**d**) and CD19+ B (**e**) cell compositions of one representative mouse from normal ($n = 11$), *SRα-tTA MYC^ON* ($n = 9$) and *SRα-tTA MYC^OFF* (doxycycline 96 h, $n = 10$) groups. **f–i** Quantification of absolute counts of total WBC (**f**), immature CD3+ CD8+ CD4+ DP lymphoma T cells (**g**), CD3−NKp46+ mature NK (**h**) and CD19+ B (**i**) cells in normal ($n = 11$), *SRα-tTA MYC^ON* ($n = 9$) and *SRα-tTA MYC^OFF* (doxycycline 96 h, $n = 10$) mice subjected to CyTOF. **j** Median fluorescence intensity (MFI) of NKp46 on splenic NK cells from normal ($n = 11$), *SRα-tTA MYC^ON* ($n = 9$) and *SRα-tTA MYC^OFF* (doxycycline 96 h, $n = 10$) mice. **k** Schematic depicting the reversible phenomenon of NK cell (blue) suppression in MYC-driven lymphomas (red). *p*-values have been calculated using a two-sided Mann–Whitney test. ns not significant. For all box plots, the center is at the median, minima and maxima are indicated by whiskers, and the box represents data between the 25th and 75th percentile.

from normal, $SR\alpha$-$tTA$ $MYC^{ON}$ (lymphoma) and $SR\alpha$-$tTA$ $MYC^{OFF}$ (regressed lymphoma) mice. Concordant with CyTOF, CIBERSORT demonstrated a reduction in the relative proportion of NK cells in $MYC^{ON}$ mice when compared to normal and $MYC^{OFF}$ mice (Supplementary Fig. 6). Hence, using three different platforms, we demonstrated that MYC causally and reversibly reduces splenic NK cell-mediated immune surveillance during lymphomagenesis (Fig. 1k).

**NK cell suppression during MYC-driven lymphomagenesis is systemic in nature.** We examined whether NK suppression occurs in other lymphoid tissues infiltrated by T-lymphoblasts, including blood and bone marrow. Circulating NK percentages, numbers per μl of blood and MFI of NKp46 were significantly reduced in overt lymphoma mice ($MYC^{ON}$), as compared to normal and MYC-inactivated ($MYC^{OFF}$) mice (Fig. 2a–e). Like that observed in blood, $CD3^-NKp46^+$ NK cells were significantly suppressed in bone marrow of lymphoma-bearing mice ($MYC^{ON}$), as compared to normal and $MYC^{OFF}$ mice (Supplementary Fig. 7e, f).

**NK cell suppression during lymphomagenesis is MYC-dependent.** To demonstrate the dependency of NK suppression on MYC, we compared percentages and surface NKp46 expression of splenic and circulating NK cells in $MYC^{OFF}$ mice with residual blasts alongside lymphoma-bearing mice ($MYC^{ON}$) with similar blast percentages (Fig. 2f, i). The presence of residual lymphoblasts in $MYC^{OFF}$ mice may be attributed to either initial high disease burden before inactivating $MYC$, or partial differentiation of T-lymphoblasts after MYC inactivation[24]. $MYC$ inactivation increases NK percentages (Fig. 2g, j) and the surface NKp46 (Fig. 2h, k) even in the presence of residual lymphoblasts. Hence, we infer that NK suppression is MYC-dependent, and does not occur simply due to the creation of space vacated by the clearance of lymphoblasts following $MYC$ inactivation.

**NK maturation is arrested during MYC-driven T-lymphomagenesis.** We investigated whether reduction in NK cells during MYC-induced lymphomagenesis occurs because of increased cell death. Surprisingly, we observed reduced NK cell death in lymphoma-bearing $MYC^{ON}$ mice, in comparison to normal and $MYC^{OFF}$ cohorts (Supplementary Fig. 8). Hence, we conclude that suppression of NK subset during lymphomagenesis is not caused by increased apoptosis of NK cells induced by lymphoma-intrinsic MYC.

The disappearance of NK cells from the bone marrow of $MYC^{ON}$ mice suggested that MYC-driven lymphomagenesis might abrogate the replenishment of mature NK cells in the periphery possibly by blocking early NK cell development in the bone marrow[25–28]. Therefore, we compared the numbers of total NK (lin$^-$CD122$^+$), NK cell precursors (NKP, lin$^-$CD122$^+$ NKp46$^-$CD49b$^-$), the more differentiated immature NK (iNK, lin$^-$CD122$^+$NKp46$^+$CD49b$^-$), and mature NK (mNK, lin$^-$CD122$^+$NKp46$^+$CD49b$^+$) cells in bone marrows of normal, $SR\alpha$-$tTA$-$MYC^{ON}$ (lymphoma), and $SR\alpha$-$tTA$-$MYC^{OFF}$ (regressed lymphoma) mice. We observed reduction in numbers of total NK, NKP, iNK and mNK cells in the bone marrow of $MYC^{ON}$ mice as compared to normal and $MYC^{OFF}$ groups (Fig. 3a–d), leading us to speculate that MYC-driven lymphomagenesis induces a block early in NK development before the NKP stage in the bone marrow. Examining the relative distributions of lin$^-$CD122$^+$ NK cells within the bone marrow, we observed that mNK percentages in the $MYC^{ON}$ mice were below the homeostatic frequencies of mNK cells present in normal and $MYC^{OFF}$ which were also mirrored in the spleen (Fig. 3e–g). These findings suggest that the

block in early NK development in the bone marrow results in a reduced supply of mNK cells that can be exported to peripheral organs including spleen.

mNK cells in the spleen undergo further differentiation to produce functional NK effectors in a 4-stage linear differentiation pathway: CD27$^-$CD11b$^-$, CD27$^+$CD11b$^-$, CD27$^+$CD11b$^+$ and CD27$^-$CD11b$^+$ [29,30]. We therefore examined whether residual mNK cells in $MYC^{ON}$ mice exhibit perturbations in the stages of terminal NK differentiation. Interestingly, we observed significant reduction in the frequencies of the least mature CD27$^-$CD11b$^-$ fraction accompanied by concomitant increases in the most terminally differentiated CD27$^-$CD11b$^+$ mNK fraction in $MYC^{ON}$ mice as compared to normal and $MYC^{OFF}$ mice (Fig. 3h, i). Hence, we conclude that the homeostatic frequencies of functional NK effector subsets are significantly impaired during MYC-driven T-lymphomagenesis. We speculate that the increase in frequency of the terminally differentiated CD27$^-$CD11b$^+$ mNK fraction in $MYC^{ON}$ mice may imply a loss in NK cell functionality in addition to the reduction in numbers due to block in early NK development. Figure 3j shows changes that occur in early (bone marrow), and late (splenic) NK maturation during MYC-driven lymphomagenesis.

Surprisingly, we observed increased frequencies of the CD27$^-$CD11b$^-$ and CD27$^+$CD11b$^-$, and decreased frequencies of CD27$^+$CD11b$^+$ and CD27-CD11b$^+$ populations in control mice as compared to that shown in the literature[29]. We found this is a result of strain-mediated differences between $C57BL/6J$[29], and $FVB/N$ strains (background for $SR\alpha$-$tTA/Tet$-$O$-$MYC$ mice (Supplementary Fig. 9)).

To see whether the observed changes in NK maturation post MYC inactivation were not a result of doxycycline treatment as described previously[31], $FVB/N$ mice were treated with doxycycline doses identical to that required to inactivate MYC expression. NK cells were measured in bone marrow and spleen by flow cytometry. No significant changes were seen between doxycycline-treated and untreated control mice in any early NK subset (Supplementary Fig. 10a–h) or the expression of CD27 and CD11b on splenic mNK cells (Supplementary Fig. 10i, j). Therefore, changes in NK homeostasis after MYC modulation in $SR\alpha$-$tTA/Tet$-$O$-$MYC$ mice are a result of changes in MYC expression and not induced by doxycycline.

To investigate whether the observed suppression of NK maturation by MYC is confined to discrete lymphoid tissue microenvironments, we examined NK cell maturation in the liver and lung of $MYC^{ON}$ mice. We observed infiltration of T-lymphoblasts in liver and lung of $MYC^{ON}$ mice accompanied by reduction in percentages of total lin$^-$CD122$^+$ NK cells, and reduced mNK cell frequency within the lin$^-$CD122$^+$ NK fraction (Supplementary Figs. 11 and 12a–d), when compared to normal mice. We also observed perturbations in the differentiation of mNK cells into functional effectors in $MYC^{ON}$ mice when compared to healthy mice (Supplementary Fig. 12e, f). Hence, NK cell maturation is perturbed in both lymphoid and non-lymphoid tissues that exhibit infiltration of T-lymphoblasts. These findings strengthen our hypothesis that observed changes in NK surveillance during MYC-driven lymphomagenesis may be directly induced by MYC expression in lymphoblasts.

**Suppression of Type I IFN by MYC may block NK surveillance.** We next investigated the molecular mechanisms behind MYC-mediated suppression of NK surveillance during primary T- and B-lymphomagenesis, by analyzing transcriptomic profiles of bulk splenic cells from normal, $SR\alpha$-$tTA$ $MYC^{ON}$ (lymphoma) and $SR\alpha$-$tTA$ $MYC^{OFF}$ (regressed lymphoma) mice (GSE106078). We observed that expression profiles of $MYC^{OFF}$ spleens

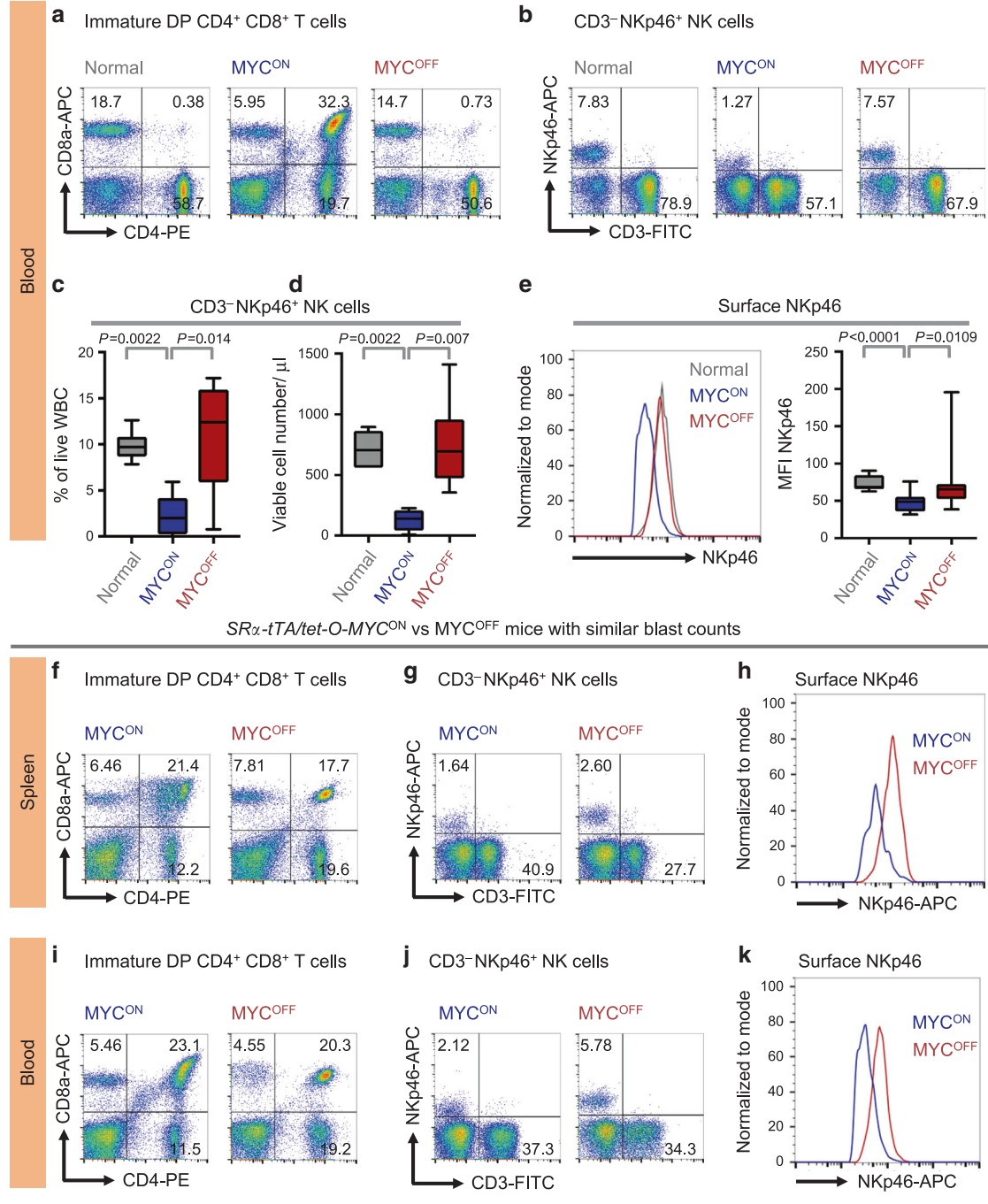

**Fig. 2 NK cell suppression in MYC-driven lymphomas is systemic and depends on MYC status. a**, **b** Representative flow cytometry plots depicting percentages of circulating immature CD4+CD8+ T (**a**) and CD3−NKp46+ NK (**b**) cells in normal ($n = 6$), SRα-tTA MYC$^{ON}$ ($n = 6$) and SRα-tTA MYC$^{OFF}$ (doxycycline 96 h, $n = 6$) mice. **c**, **d** Quantification of percentages (**c**) and absolute counts per μl of blood (**d**) of CD3−NKp46+ NK cells, in normal ($n = 6$), SRα-tTA MYC$^{ON}$ ($n = 6$), and SRα-tTA MYC$^{OFF}$ (doxycycline 96 h, $n = 6$) mice. **e** Comparison of MFI of surface NKp46 on circulating NK cells in normal ($n = 6$), SRα-tTA MYC$^{ON}$ ($n = 6$) and SRα-tTA MYC$^{OFF}$ (doxycycline 96 h, $n = 6$) mice. **f**, **g** Representative flow cytometry plots depicting percentages of splenic immature CD4+CD8+ T (**f**) and CD3−NKp46+ NK (**g**) cells in SRα-tTA MYC$^{ON}$ and SRα-tTA MYC$^{OFF}$ (doxycycline 96 h) mice with comparable blast counts. **h** Comparison of surface NKp46 levels (MFI) on splenic NK cells from SRα-tTA MYC$^{ON}$ and SRα-tTA MYC$^{OFF}$ (doxycycline 96 h) mice with comparable blast counts. **i**, **j** Representative flow cytometry plots depicting percentages of circulating immature CD4+CD8+ T (**i**), and CD3−NKp46+ NK (**j**) cells in SRα-tTA MYC$^{ON}$ and SRα-tTA MYC$^{OFF}$ (doxycycline 96 h) mice with comparable blast counts. **k** Comparison of surface NKp46 levels (MFI) on circulating NK cells from SRα-tTA MYC$^{ON}$ and SRα-tTA MYC$^{OFF}$ (doxycycline 96 h) mice with comparable blast counts. p-values have been calculated using a two-sided Mann–Whitney test. ns not significant. For all box plots, the center is at the median, minima and maxima are indicated by whiskers, and the box represents data between the 25th and 75th percentile.

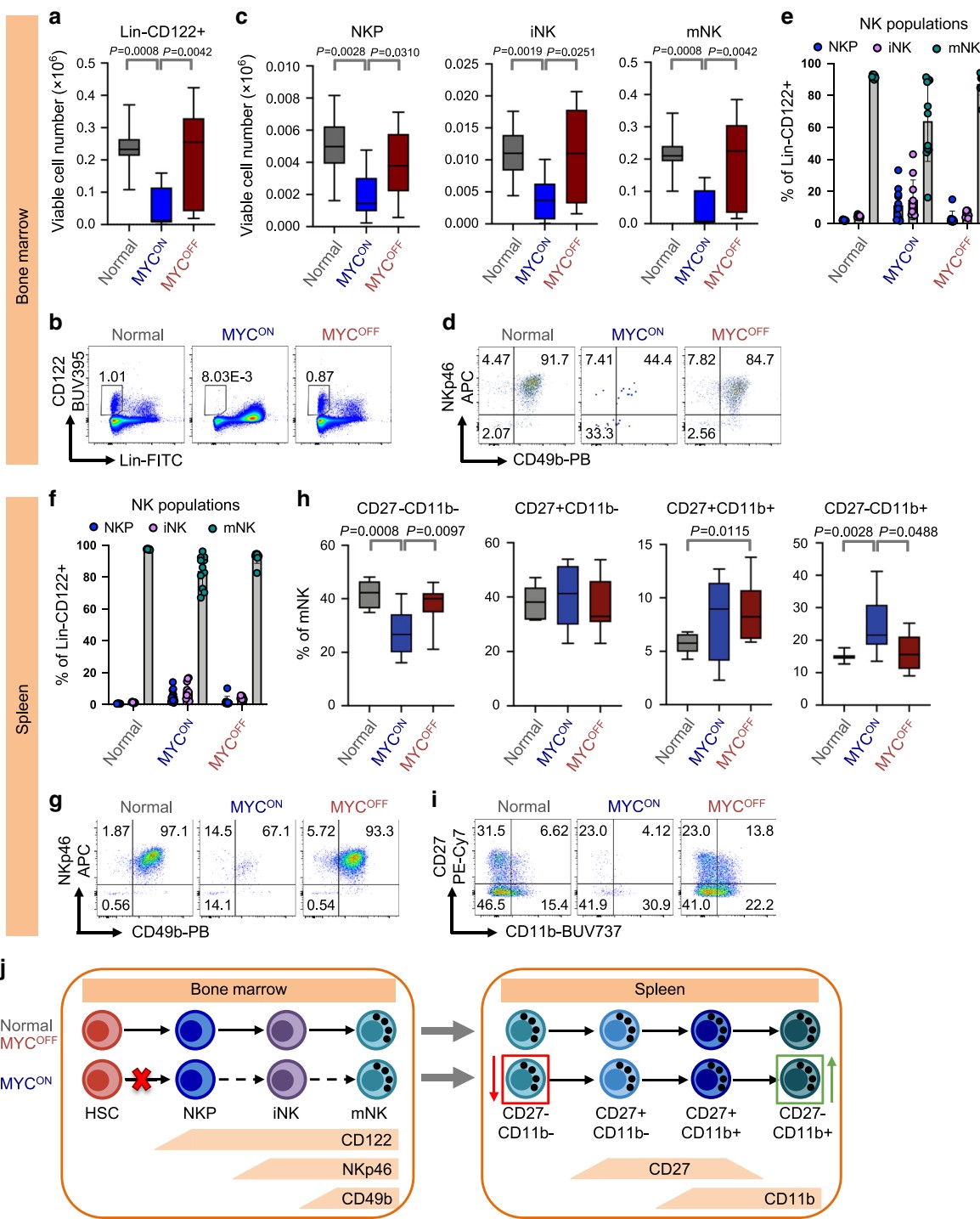

**Fig. 3 Maturation of NK cells is arrested in mice bearing overt MYC-driven T-lymphomas. a**, **b** Quantification of viable cell numbers (**a**) representative flow cytometry plots (**b**) of Lin−CD122+ in bone marrow from normal (n = 7), SRα-tTA MYCON (n = 11) and SRα-tTA MYCOFF (doxycycline 96 h, n = 9) measured by flow cytometry. **c**–**e** Quantification of viable cell numbers (**c**), representative flow cytometry plots (**d**) and percentages (**e**) of NK precursors (NKP, Lin−CD122+NKp46−CD49b−), immature NK cells (iNK, Lin−CD122+NKp46+CD49b−) and mature NK cells (mNK, Lin−CD122+NKp46+CD49b+) in bone marrow from normal (n = 7), SRα-tTA MYCON (n = 11) and SRα-tTA MYCOFF (doxycycline 96 h, n = 9). Data in **e** show mean + s.d. **f**, **g** Quantification of percentages (**f**) and representative flow cytometry plots (**g**) of NKP, iNK and mNK in spleen from normal (n = 7), SRα-tTA MYCON (n = 11) and SRα-tTA MYCOFF (doxycycline 96 h, n = 9). Data in **f** show mean + s.d. **h**, **i** Quantification of percentages (**h**) and representative flow plots (**i**) of mNK cells expressing CD27 and CD11b in spleen from normal (n = 7), SRα-tTA MYCON (n = 11) and SRα-tTA MYCOFF (doxycycline 96 h, n = 9). Populations are gated on Lin−CD122+ NKp46+CD49b+ intact live singlets. **j** Model depicting the arrest in differentiation from HSC to NKP stage in bone marrow and the changes in mature splenic NK effector subsets during MYC-driven lymphomagenesis. Arrested step in NK cell differentiation is indicated by a red cross and dashed arrows indicate stages of NK cell differentiation that do not occur because of the arrest. The red and green boxes indicate NK effector subsets whose frequencies are reduced and enhanced respectively, during MYC-driven lymphomagenesis. p-values have been calculated using a two-sided Mann–Whitney test. For all box plots, the center is at the median, minima and maxima are indicated by whiskers, and the box represents data between the 25th and 75th percentile.

clustered closely with normal spleens, thus confirming that transcriptional changes induced by MYC were largely reversible upon its inactivation (Fig. 4a). Gene set enrichment analyses (GSEA) revealed that the JAKs (Janus Kinases)—STAT1/2 (Signal Transducer and Activator of Transcription 1/2)— Type I IFN (Interferon) pathway was significantly suppressed in $MYC^{ON}$ mice when compared to normal and $MYC^{OFF}$ mice (Fig. 4b–d).

Notably, the activation of STAT1/2 post *MYC* inactivation resembles an antiviral immune response associated with production of Type I IFNs and subsequent activation of NK cell-mediated immune surveillance[32]. STAT1/2-Type I IFN signaling has been demonstrated to impact maturation and function of NK cells[33–39]. Compared to normal counterparts, mice deficient in Type I IFN receptor ($IFN\alpha R1/2^{-/-}$) show significantly lower splenic mNK cell frequencies[33,36], a block in early NK maturation in the bone marrow before NKP stages[35], and altered frequencies of mNK subsets in the NK effector differentiation pathway marked by changes in CD27 and CD11b[36]. We therefore speculated that reduced NK surveillance in $MYC^{ON}$ mice may phenocopy mice deficient in components of the Type I IFN signaling cascade.

Concordant with NK suppression during MYC-driven lymphomagenesis, we observed transcriptional repression of many STAT1/2-Type I IFN regulated genes that are specific to NK-mediated immune surveillance in $MYC^{ON}$ mice, when compared to normal and $MYC^{OFF}$ mice (Supplementary Fig. 13). This suggested the existence of a link between Type I IFN signature and the observed changes in splenic NK cell maturation.

Next, we compared transcript levels of *STAT1*, *STAT2* and other Type I IFN signaling components in normal B cells ($MYC^{LOW}$), pre-lymphoma B cells from *Eμ-MYC* mice ($MYC^{HIGH}$), and overt B-cell lymphoma from *Eμ-MYC* mice ($MYC^{HIGH}$), a mouse model of MYC-driven B-lymphoma (GSE51011[40]). Sequential increase in MYC during stepwise B-cell transformation was accompanied by transcriptional suppression of STAT1/2-Type I IFN signature (Supplementary Fig. 14). Hence, suppression of STAT1/2-Type I IFN signaling is a signature of both T- and B- MYC-driven lymphomas.

**MYC-driven suppression of NK cells is mediated by Type I IFN signaling**. We investigated whether short-term (3 days) administration of Type I IFNα to *SRα-tTA MYC$^{ON}$* mice bearing overt T-lymphoma can rescue MYC-mediated NK suppression. The percentages and numbers of $CD3^-NKp46^+$ NK cells were significantly increased in IFNα-treated mice when compared to litter-matched vehicle-treated lymphoma mice (PBS) (Fig. 4e). Of note, numbers of WBC, $CD3^+$ T cells and DP immature T cells remain unaltered between the control and IFNα-treated groups (Supplementary Fig. 15), suggesting that Type I IFNs specifically regulate NK cell homeostasis. Long-term administration of IFNα to $MYC^{ON}$ overt T-lymphoma mice improves overall survival (OS) in comparison to vehicle (PBS)-treated lymphoma-bearing mice (Fig. 4f).

To address whether depleting NK cells during IFNα administration impacted OS, we treated overt T-lymphoma-bearing *SRα-tTA MYC$^{ON}$* mice with either anti-NK1.1 NK cell-depleting antibody or IgG2a control antibody in combination with recombinant IFNα. We observed that depleting NK cells shortened OS time in contrast to controls where NK cells were not depleted (Fig. 4g). Hence, we conclude that anti-tumor effects of Type I IFN are at least in part dependent on NK cells. These findings corroborate our earlier observations that Type I IFN exerts its anti-tumor effects by not directly acting on the T-lymphoblasts but rather on the host immune cells.

Finally, we interrogated the requirement of Type I IFN signaling in restoring homeostatic frequencies of NK fractions during MYC inactivation by treating overt T-lymphoma-bearing $MYC^{ON}$ mice with IFNAR1 blocking or control antibody at the time of MYC inactivation, and examining splenic NK maturation after 4 days. We observed no differences in the numbers of total splenic $lin^-CD122^+$ NK cells, and the relative frequencies of NKP, iNK and mNK fractions within the $lin^-CD122^+$ subset between IFNAR1 blocking antibody-treated and control-treated groups (Supplementary Fig. 16). However, upon examining the frequencies of splenic mNK subsets, we observed a significant decrease in the frequency of $CD27^+CD11b^+$ effector cells upon blocking IFNAR1 during MYC inactivation (Fig. 4h, i). This indicates that albeit Type I IFN signaling is required for generating functional NK effectors during late stages of NK maturation, it is dispensable for early NK maturation and restoration of normal NK numbers post MYC inactivation. We conclude that MYC-driven lymphomagenesis may subvert normal immunological processes of NK cell homeostasis by suppressing STAT1/2 signaling cascade and Type I IFN production.

**Oncogenic MYC suppresses Type I IFN production by tumor cells**. Next, we investigated whether modulation of MYC in vitro in MYC-driven B- and T- lymphoma cell lines differentially regulates STAT1/2-Type I IFN signaling. As Burkitt's Lymphoma (BL) is a classic human MYC-associated lymphoma, we measured signaling changes by Phospho-CyTOF before ($MYC^{ON}$) and after ($MYC^{OFF}$) MYC inactivation in an EBV-transformed human B-cell line that expresses MYC regulated by the Tet System (P493-6), and mimics BL[41]. We confirmed that *MYC* is inducibly expressed in P493-6 cells (Fig. 5a). Amongst immunoregulatory cytokine signaling pathways that operate through the JAKs and STATs, inactivation of *MYC* in P493-6 cells increased expression and activation of STAT1 compared to $MYC^{ON}$ P493-6 cells (Fig. 5b–d, Supplementary Figs. 17, 18 and 19a). Transcription of *STAT2*, the binding partner of STAT1, was induced post MYC inactivation in P493-6 cells (Fig. 5d, Supplementary Fig. 19b). Indeed, the transcript level of Type I *IFNα2* was significantly elevated after MYC inactivation in P493-6 cells ($MYC^{OFF}$, Fig. 5e). Cytokine profiling of the P493-6 cells pre- and post MYC inactivation demonstrated that the secretion of IFNα is significantly upregulated in $MYC^{OFF}$ cells in comparison to $MYC^{ON}$ cells (Fig. 5f, Supplementary Fig. 20).

Tumor-intrinsic signaling changes post MYC inactivation in our T-lymphoma line derived from *SRα-tTA/tet-O-MYC* mice (Supplementary Fig. 21) mirrored those observed in the human BL-like model with MYC inactivation leading to activation of STAT1/2-Type I IFN signaling (Supplementary Figs. 22 and 23). Our findings suggested that lymphoma-intrinsic MYC represses STAT1/2-Type I IFN signaling; thereby blocking the maturation of NK cells in the tumor microenvironment.

**MYC transcriptionally represses STAT1/2-Type I IFN signaling**. We tested whether MYC transcriptionally represses the STAT1/2-Type I IFN signaling cascade. Although widely known as a transactivator, MYC can inhibit gene transcription by sequestering transactivators including MIZ1 and Sp1/Sp3[42–44], and binding to Initiator (Inr) elements in gene promoters. For example, MYC inactivation releases MYC-MAX heterodimers from the binding sites of tumor suppressor genes such as *Cdkn2b* and *Cdkn1b*, thereby promoting MIZ1-mediated transcriptional activation of these genes[42–44].

Concordant with a previous study where MYC was shown to transcriptionally repress *STAT1*[45], we found that MYC binds the

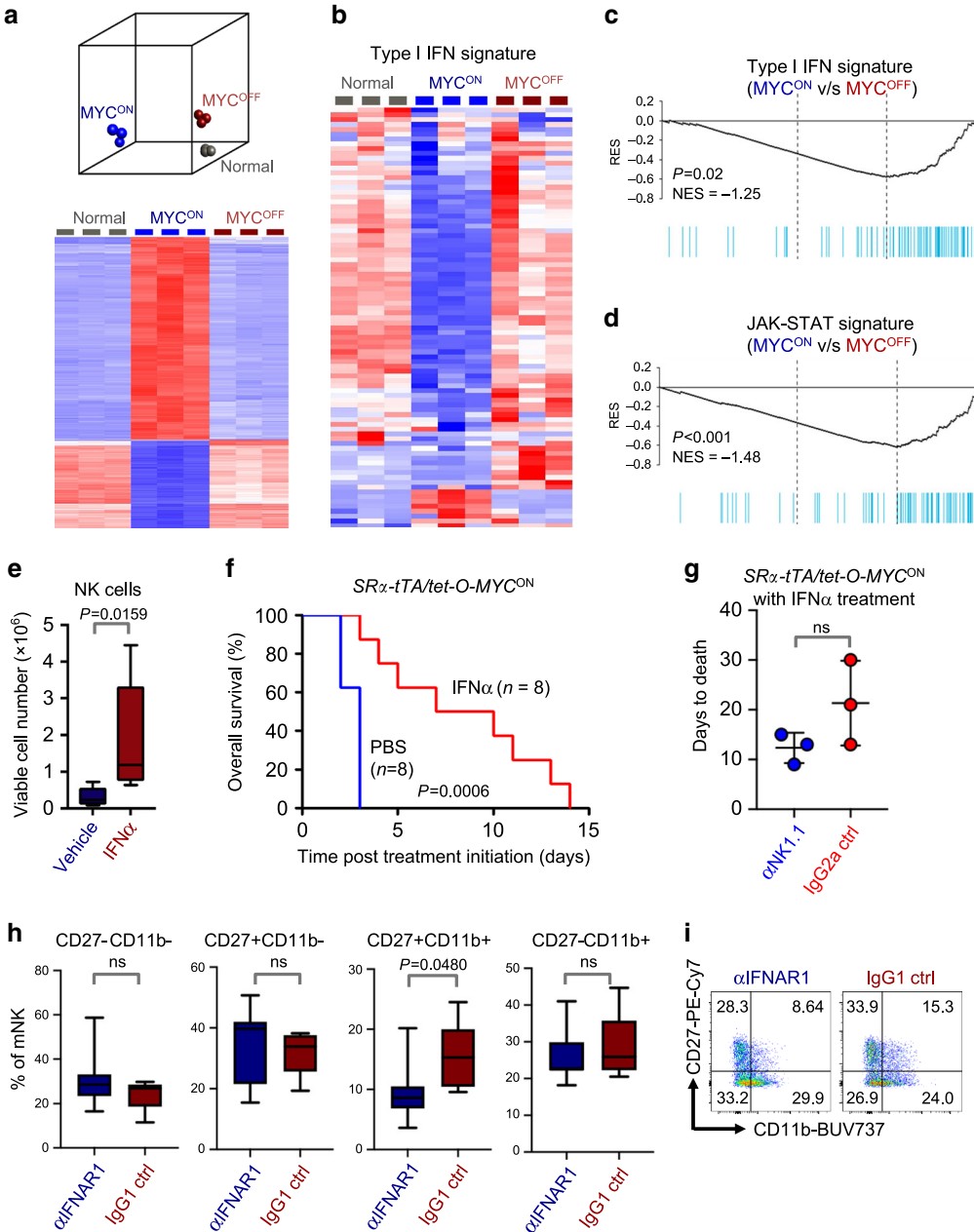

**Fig. 4 Suppression of Type I IFN signaling mediates escape from NK surveillance during MYC-driven lymphomagenesis. a** Principal component analysis (PCA, top) and heatmap (bottom) depicting global transcriptomic profiles of bulk splenic samples derived from normal ($n = 3$), $SR\alpha$-tTA MYC$^{ON}$ ($n = 3$) and $SR\alpha$-tTA MYC$^{OFF}$ ($n = 3$) mice, obtained by RNA sequencing (GSE106078). **b** Heatmap showing suppression of Type I IFN signaling in spleens of $SR\alpha$-tTA MYC$^{ON}$ ($n = 3$) mice when compared to normal ($n = 3$) and $SR\alpha$-tTA MYC$^{OFF}$ ($n = 3$) mice. **c** Gene set enrichment analysis (GSEA) depicting suppression of Type I IFN signature in spleens of $SR\alpha$-tTA MYC$^{ON}$ ($n = 3$) mice when compared to $SR\alpha$-tTA MYC$^{OFF}$ ($n = 3$) mice. **d** GSEA depicting suppression of JAK-STAT signature in spleens of $SR\alpha$-tTA MYC$^{ON}$ ($n = 3$) mice when compared to $SR\alpha$-tTA MYC$^{OFF}$ ($n = 3$) mice. **e** Quantification of absolute counts of CD3$^-$ NKp46$^+$ NK cells in $SR\alpha$-tTA/tet-O-MYC mice bearing overt lymphomas ($SR\alpha$-tTA MYC$^{ON}$) treated for 3 days with either IFN$\alpha$ ($n = 5$), or vehicle (PBS, $n = 5$). **f** Comparison of overall survival (OS) probabilities between PBS ($n = 7$) and IFN$\alpha$ ($n = 7$)-treated $SR\alpha$-tTA/tet-O-MYC mice bearing overt lymphomas ($SR\alpha$-tTA MYC$^{ON}$, age = 2–3 months). **g** Comparison of time to morbidity in $SR\alpha$-tTA/tet-O-MYC mice bearing overt lymphomas ($SR\alpha$-tTA MYC$^{ON}$) treated with IFN$\alpha$ combined with either anti-NK1.1 ($n = 3$) or IgG2a control ($n = 3$). Data show mean ± s.d. **h, i** Quantification of percentages (**h**) and representative flow cytometry plots (**i**) of mNK cells expressing CD27 and CD11b in spleen of SR$\alpha$-tTA MYC$^{OFF}$ (doxycycline 96 h) treated with anti-IFNAR1 ($n = 7$) or IgG1 control ($n = 5$). p-values have been calculated using a two-sided Mann–Whitney test. p-values: ns not significant. For all box plots, the center is at the median, minima and maxima are indicated by whiskers, and the box represents data between the 25th and 75th percentile.

*STAT1* promoter in P493-6 human BL and mouse MYC-driven T-lymphoma (Supplementary Fig. 24). To delineate the molecular mechanism behind MYC-mediated transcriptional repression of Type I IFN signaling, we compared levels of *STAT1*, *STAT2* and *IFNα2* in MYC-driven lymphoma cell lines overexpressing MYC, or a MYC mutant that fails to sequester MIZ1 (MYC V394D), or the corresponding empty vector (EV). Although MYC overexpression in P493-6 cells repressed the production of STAT1/2

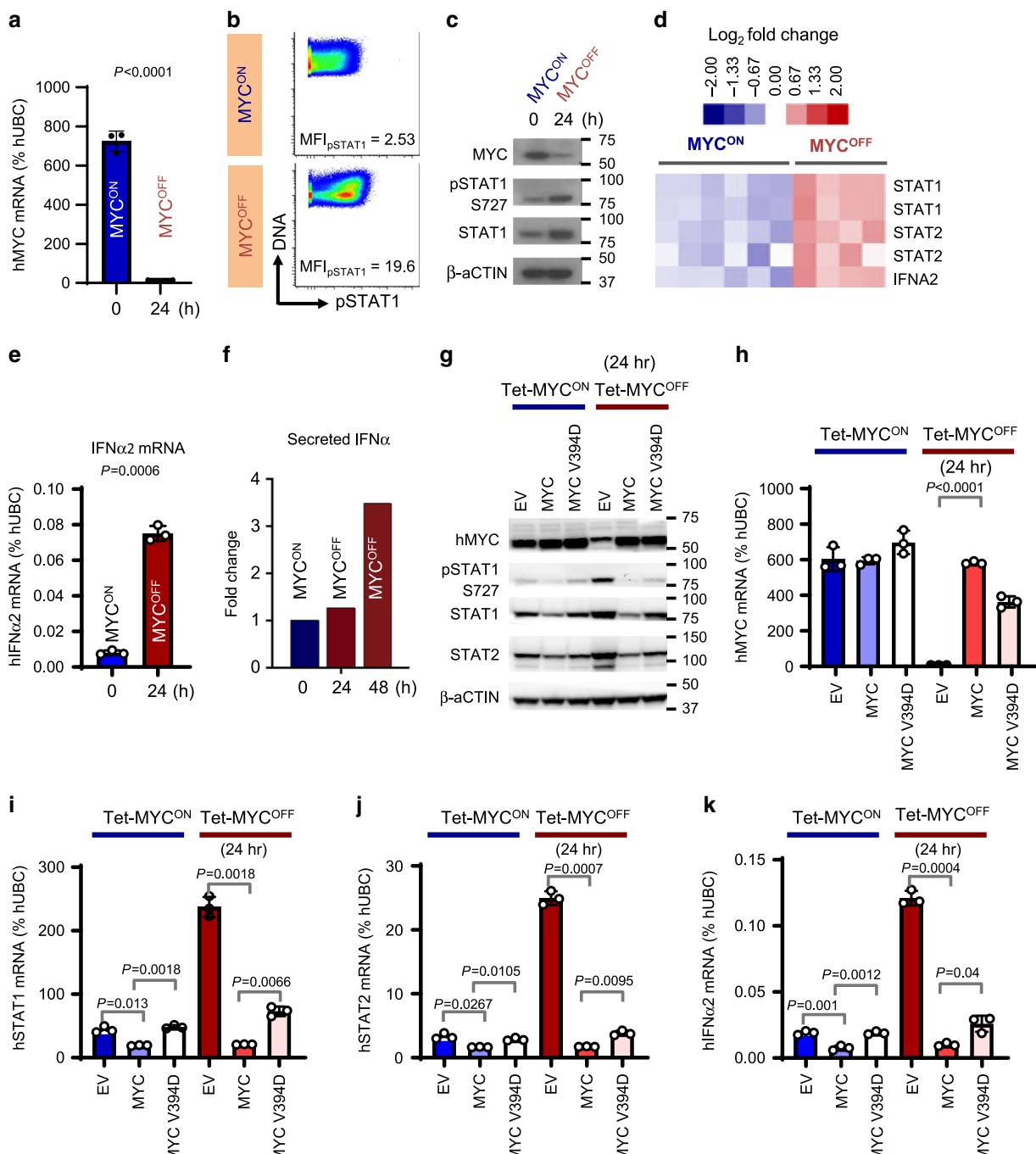

**Fig. 5 MYC transcriptionally represses the STAT1/2-Type I IFN signaling associated with NK cell maturation. a** Quantitative real-time PCR for MYC in P493-6 BL human cells before and after MYC inhibition for 24 h by doxycycline (*n* = 3, mean ± s.d.). **b** Phospho mass cytometry (Phospho-CyTOF) depicting changes in pSTAT1 levels before (*MYC^ON^*) and after (*MYC^OFF^*) MYC inactivation for 24 h in P493-6 cells. **c** Immunoblotting to validate changes in pSTAT1 observed by phospho-CyTOF. **d** Heatmap depicting transcriptional changes in STAT1/2-Type I IFN signaling cascade before (*MYC^ON^*, *n* = 6) and after MYC inactivation (*MYC^OFF^*, 24 h, *n* = 4) in P493-6 cells using expression arrays. **e** Quantitative real-time PCR for Type I IFNα2 in P493-6 BL human cells before and after MYC inactivation for 24 h by doxycycline (*n* = 3, mean ± s.d.). **f** Luminex assay depicting changes in Type I IFNα2 secreted from P493-6 cells before and after MYC inactivation for 24 and 48 h. Each time point was carried out in duplicates. Average of these duplicates is shown. **g** Immunoblotting in P493-6 BL cell lines expressing empty vector (EV), *MYC* overexpression vector (MYC), or a MYC mutant that fails to bind MIZ1 (*MYC V394D*), pre- (*Tet-MYC^ON^*) and post (*Tet-MYC^OFF^*) MYC inactivation. **h–k** Quantitative real-time PCR for MYC (**h**), STAT1 (**i**), STAT2 (**j**) and Type I IFNα2 (**k**) in P493-6 BL cell lines expressing empty vector (EV), MYC overexpression vector (MYC), or a MYC mutant that fails to bind MIZ1 (MYC V394D), pre- (*Tet-MYC^ON^*) and post (*Tet-MYC^OFF^*) MYC inactivation (*n* = 3, mean ± s.d.). All *p*-values have been calculated using two-tailed Student's *t*-test. ns not significant.

and IFNα2 (Fig. 5g–k, Supplementary Fig. 25), the MYC V394D mutant significantly rescued the activation of STAT1/2-Type I IFN signaling, both before and after inactivation of tetracycline-controlled MYC (Tet-MYC). Our results show that MYC transcriptionally represses STAT1/2-Type I IFN signaling partially by sequestering transcriptional activators such as MIZ1. In MYC-driven T-lymphoblasts, we observe that MYC-mediated suppression of the Type I IFN signaling cascade is independent of MIZ1 (Supplementary Figs. 26 and 27), suggesting the possible involvement of other MYC binding partners such as SP1/3[40].

**MYC^High human lymphomas suppress STAT1/2 and NK surveillance.** We examined MYC-driven human lymphoma patients for evidence of repression of STAT1/2 signaling and NK cell-mediated immune surveillance. First, we analyzed human lymphomas that often exhibit MYC hyperactivation, such as Burkitt's Lymphoma (BL) and Diffuse Large B-Cell Lymphoma (DLBCL, GSE4475[46]). We observed that STAT1 and STAT2 transcripts inversely correlate with MYC transcript levels in BL and DLBCL patients (Fig. 6a, b). Of note, BL, which is driven by MYC translocations[47] displays the highest MYC levels in combination with the lowest levels of STAT1 and STAT2. This indicates that natural addiction to MYC in BL might lead to the suppression of STAT1/2-Type I IFN signaling.

Next, we analyzed whether separation of BL patients into four groups based on their median expressions of both STAT1 and STAT2 can predict clinical prognosis. We observed that patients with the lowest levels of both STAT1 and STAT2 had the most unfavorable clinical outcome (Fig. 6c). Furthermore, the majority (85%, 6/7) of BL patients with lowest levels of both STAT1 and STAT2 have no activated NK cells (Fig. 6d), estimated by CIBERSORT. We found that higher levels of activated NK cells in the BL patients were associated with a significantly favorable clinical outcome and prolonged survival (Fig. 6e). Hence, low expression of both STAT1 and STAT2 coincides with both NK cell suppression and poor overall survival in BL patients.

Finally, using CIBERSORT on bulk expression data, we compared the estimated fractions of NK cells in mononuclear cells isolated from blood of MYC^High T-lymphoma patients (GSE62156)[48,49] and healthy individuals[23]. Of note, MYC levels are significantly higher in T-lymphoma patients as compared to healthy individuals (Fig. 6f) because of chromosomal translocations that can potentially activate MYC[48–54]. We observed a significant suppression of STAT1 and STAT2 levels (Fig. 6g, h), and a reduction in total and activated NK cell subsets in T-lymphoma patients in comparison to their normal counterparts (Fig. 6i–l). Collectively, our findings suggest a common paradigm of NK suppression in both human and mouse MYC-driven lymphomas, which occurs in part due to the direct repression of the STAT1/2-Type I IFN signaling cascade by the MYC oncogene.

**NK cells as a potential therapy against MYC-driven lymphomas.** Our findings suggest that subversion of NK surveillance is critical for MYC-driven lymphomagenesis. Therefore, we investigated whether NK cells can be developed as an immunotherapy against MYC-driven lymphomas by interrogating the requirement and sufficiency of NK cells in MYC-driven lymphoma initiation, and recurrence post MYC inactivation.

To examine the requirement of NK cells in blocking lymphoma growth in transgenic MYC-driven T-lymphoma mice, we depleted NK cells in 4-week-old SRα-tTA-MYC^ON mice prior to overt lymphoma development at 8–12 weeks, and compared these to control mice. We observed that 100% of NK-depleted mice succumbed to aggressive lymphomas accompanied by splenomegaly, whereas control mice showed a heterogeneous pattern of

disease severity with only 50% of these developing splenomegaly (Supplementary Fig. 28a, c). Comparison of disease-free survival (DFS) Probabilities indicated slightly prolonged survival in the control group in comparison to the NK-depleted group albeit not statistically significant (Supplementary Fig. 28a). We predict that increasing the sample sizes to include more mice per group could lead to statistically significant changes in DFS between the two groups. Of note, we observed a statistically significant increase in the survival probabilities of control mice when morbidity was measured as a function of splenomegaly which indicates high disease severity (Supplementary Fig. 28b). Hence, we conclude that NK cells are important for blocking MYC-driven lymphomagenesis.

We further corroborated the requirement of NK cells in slowing down T-lymphoma growth by injecting intravenously luciferase-labeled MYC^ON T-lymphoblasts derived from an SRα-tTA/tet-O-MYC mouse (Supplementary Fig. 21) into recipients with (NOD SCID) or without (NOD SCID IL-2Rγ^−/−, NSG) NK and other innate lymphoid cells. Lymphoma onset (day 5 for NSG and day 18 for NOD SCID), and median morbidity (day 10 for NSG and day 20 for NOD SCID) of mice were significantly delayed in NOD SCID transplant recipients (Fig. 7a, b). T-lymphomas eventually engrafted in the bone marrow of NOD SCID mice where NK cells are present (Fig. 7a). We infer that although NK surveillance initially blocks MYC-driven lymphomagenesis, continued MYC expression eventually suppresses NK cells to establish lymphomas, as seen in primary SRα-tTA/tet-O-MYC mice.

Engraftment of T-lymphoma at sites with NK surveillance such as the bone marrow (Fig. 7a, BLI image day 18 post injection) suggested that NK cell-mediated allogeneic rejection is an unlikely explanation for delayed lymphoma engraftment in NOD SCID mice. However, to rule this out, we transplanted syngeneic mature NK cells (CD3−NKp46+) obtained from healthy normal FVB/N mice (Supplementary Fig. 29) into NSG mice 3 days before (d-3) T cell lymphoma transplantation (d0). We observed significantly delayed T-lymphoma initiation in transplant recipients that received syngeneic NK cells compared to transplant recipients that received vehicle (Fig. 7c, d). Hence, NK cells alone are sufficient to slow down the growth of MYC-driven lymphomas. Of note, despite delayed lymphoma initiation in transplant recipients that received NK cells, systemic lymphoma eventually develops consistent with our previous findings that, once established, MYC-driven lymphomas promote NK cell suppression.

MYC-driven lymphomas recur post MYC inactivation with 100% penetrance in immune-deficient hosts (NSG) in comparison to immune-competent hosts where MYC inactivation sustains lymphoma regression[10,55]. We investigated whether NK cells are sufficient to delay lymphoma recurrence post MYC inactivation. Syngeneic NK cells (CD3−NKp46+) from normal mice when adoptively-transferred into T-lymphoma-bearing NSG recipients at the time of MYC inactivation delayed lymphoma recurrence and prolonged OS (Fig. 7e–g) when compared to vehicle-treated controls. Thus, adoptive transfer of NK cells alone is sufficient to sustain lymphoma regression post MYC inactivation.

Collectively, our results suggest that NK cells have an important anti-tumorigenic role in MYC-driven lymphoms, hence making NK cell-based immune therapies an attractive treatment strategy against such malignancies.

## Discussion
Mapping perturbations in immune subsets during primary tumorigenesis is crucial for identifying cancer immunotherapies. Through CyTOF, we delineated the immune signature of MYC-

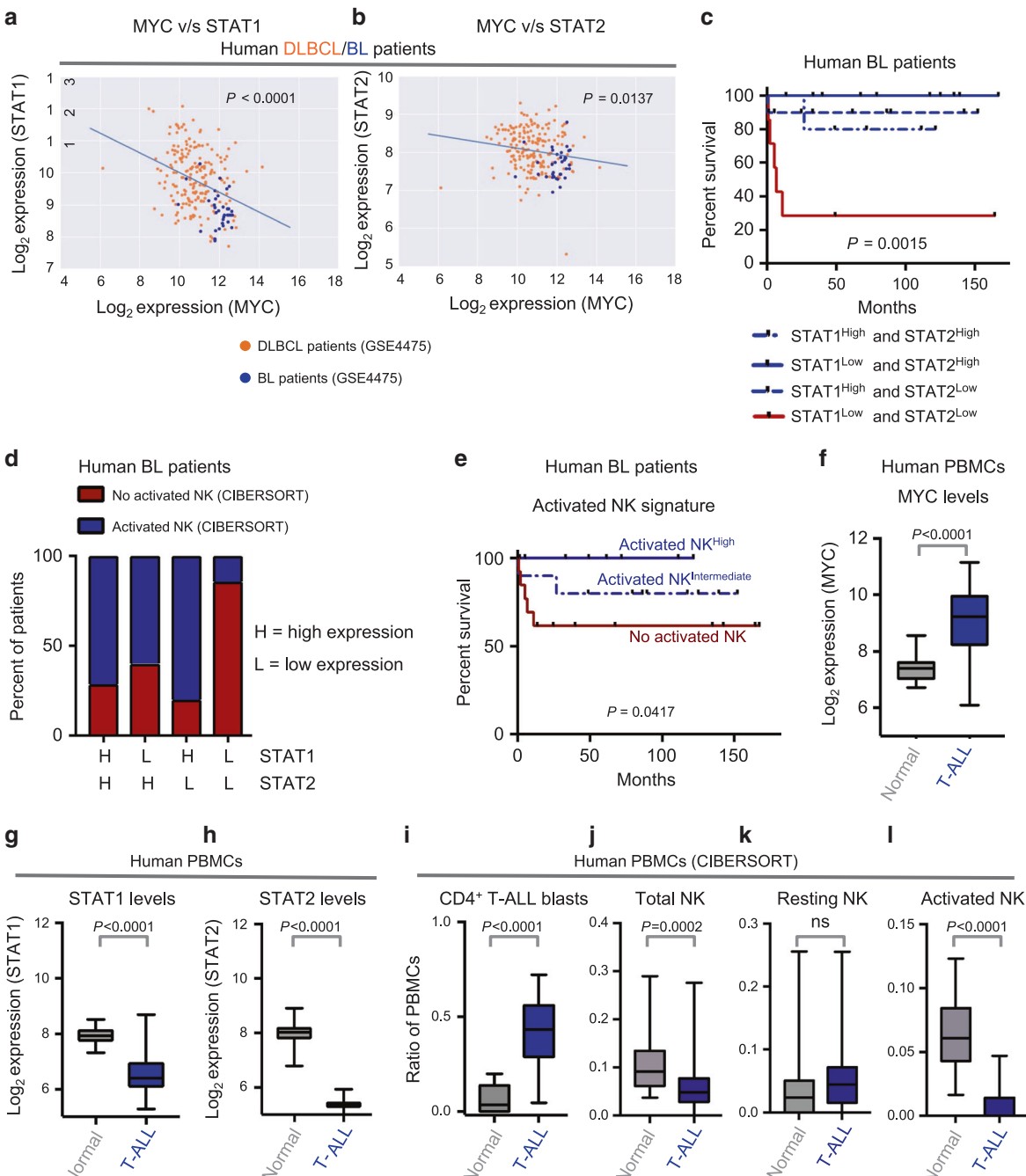

**Fig. 6 MYC^High human lymphomas exhibit reduced STAT1/2 and NK cell-mediated immune surveillance. a, b** Correlations between mRNA levels of MYC and STAT1 (**a**), or MYC and STAT2 (**b**) in classical MYC-driven human lymphoma patient samples (*n* = 187, DLBCL, orange; *n* = 34, BL, blue). **c** Multivariate analysis comparing overall survival probabilities of BL patients (*n* = 34, GSE4475) when divided into four groups based on combined levels of STAT1 and STAT2. **d** Proportions of patients with no activated NK cells (estimated by CIBERSORT) when BL patients (*n* = 34, GSE4475) are divided into four groups based on combined levels of STAT1 and STAT2. **e** Comparison of overall survival probabilities of BL patients (*n* = 34, GSE4475) separated into three categories based on levels of activated NK cells in the tumor estimated by CIBERSORT, as none (0%, no activated NK), intermediate (activated NK^Intermediate) and high (activated NK^High). **f–h** Comparison of mRNA levels of MYC (**f**), STAT1 (**g**) and STAT2 (**h**) in mononuclear cells isolated from healthy human blood (*n* = 20) and from blood of T-ALL patients (*n* = 48, GSE62156). **i–l** CIBERSORT estimates of relative proportions of T cell blasts (**i**), and NK subsets, namely, total NK (**j**), resting NK (**k**) and activated NK (**l**), in peripheral blood mononuclear cells isolated from healthy individuals (*n* = 20) and T-ALL patients (*n* = 48, GSE62156). *p*-values for all survival analyses have been calculated using the log-rank test. *p*-values for all other analyses have been calculated using a two-sided Mann–Whitney test. *p*-values: ns not significant. For all box plots, the center is at the median, minima and maxima are indicated by whiskers, and the box represents data between the 25th and 75th percentile.

driven T-lymphomas before and after *MYC* inactivation. We identified a mechanism by which *MYC* directly suppresses NK surveillance. MYC-dependent suppression of NK cells was highly specific as other anti-tumorigenic immune subsets were not altered in numbers by *MYC* modulation (Fig. 1, Supplementary Figs. 2–4). Our results are consistent with a recent report where MYC has been shown to promote NK cell exclusion in adenocarcinomas[11].

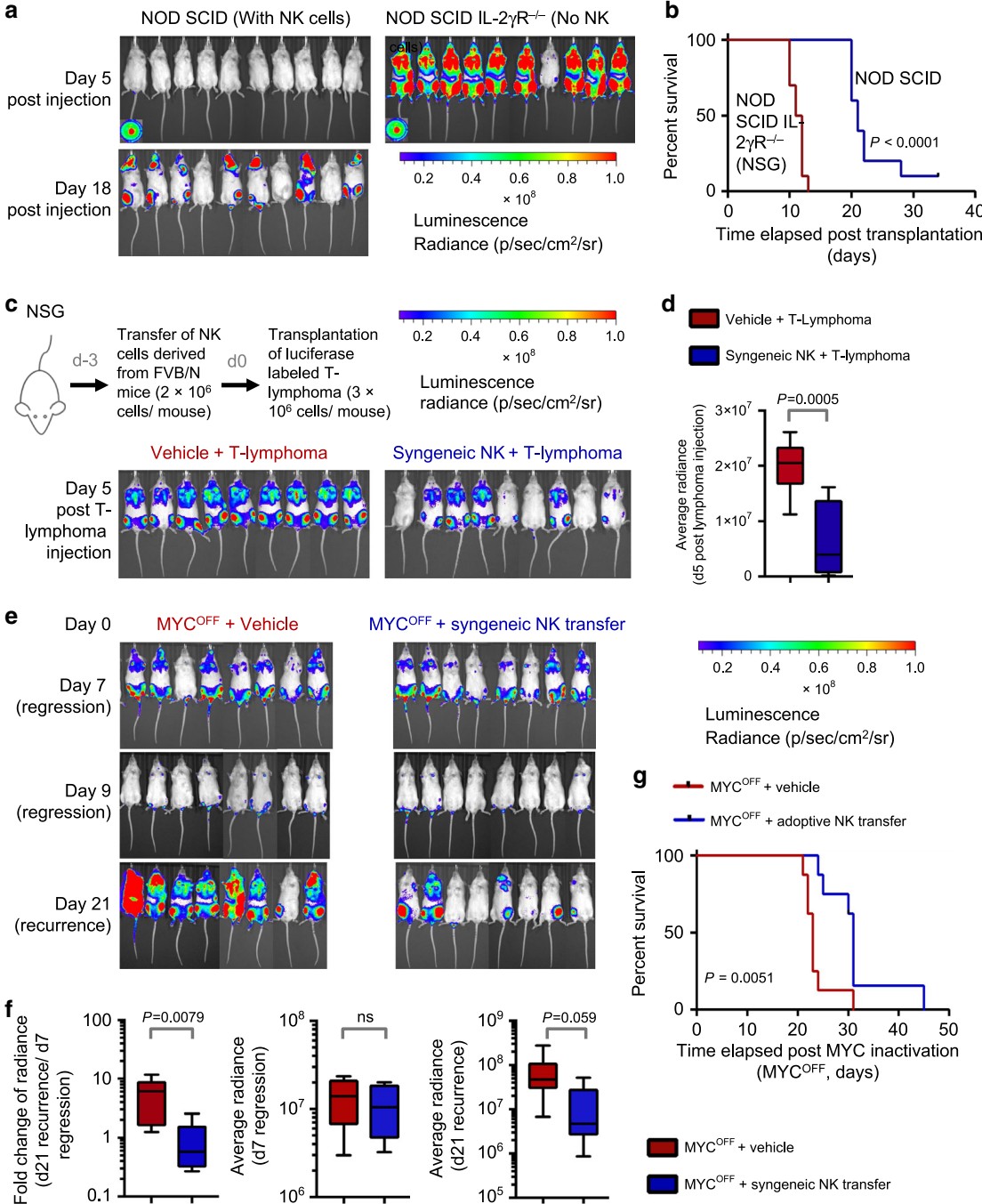

**Fig. 7 NK cells slow down growth and recurrence of MYC-driven lymphomas. a** Bioluminescence imaging (BLI) of *NOD-SCID* (*n* = 10) and *NOD-SCIDIL2Rγ*^−/− (*NSG*, *n* = 10) mice transplanted with *SRα-tTA MYC*^ON mouse T-lymphoma cells post injection to monitor lymphoma engraftment and initiation. **b** Comparison of survival probabilities of *NOD-SCID* (*n* = 10) and *NOD-SCIDIL2Rγ*^−/− (*NSG*, *n* = 10) T-lymphoma transplant recipients. **c** BLI of *NSG* T cell lymphoma transplant recipients comparing lymphoma engraftment and initiation in the presence (*n* = 9) and absence (*n* = 9) of NK cells syngeneic to the tumor. **d** Quantification of changes in BLI signal of *NSG* transplant recipients that received vehicle or syngeneic NK cells before transplantation of T-lymphoma cells. **e** BLI of *NSG* T cell lymphoma transplant recipients comparing recurrence rates upon MYC inactivation in the absence (*MYC*^OFF + vehicle, *n* = 8) or presence of NK cells (*MYC*^OFF + Adoptive NK transfer, *n* = 8). **f** Quantification of changes in BLI signal (left) and absolute signals (middle and right) at the time of lymphoma regression (d7 *MYC*^OFF) and lymphoma relapse (d21 *MYC*^OFF). **g** Comparison of survival probabilities of NSG T-lymphoma transplant recipients that received either vehicle (*n* = 8) or adoptive NK transfer (*n* = 8) at the time of MYC inactivation (*MYC*^OFF). *p*-values for all survival analyses have been calculated using the log-rank test. *p*-values for all other analyses have been calculated using a two-tailed Mann–Whitney test. ns not significant. For all box plots, the center is at the median, minima and maxima are indicated by whiskers, and the box represents data between the 25th and 75th percentile.

We found that mice bearing overt MYC-driven T-lymphomas suppress early NK cell development in bone marrow (Fig. 3a–e). We show that such an early developmental arrest in NK maturation translates to the periphery, possibly due to the reduced export of mNK cells to organs including spleen, liver and lung (Fig. 3f and Supplementary Figs. 11a and 12a–d). We show that mNK cells in $MYC^{ON}$ mice exhibit altered frequencies of the subsets of effector NK differentiation pathway[29,30], suggesting that NK function may be impaired in MYC-driven lymphomas. We found that MYC inactivation significantly reverses the defects both in early and late stages of NK development observed in $MYC^{ON}$ mice (Fig. 3h). Hence, we infer that MYC-driven lymphomas evade NK surveillance by blocking NK cell maturation, and inhibiting MYC is essential for reversing this developmental blockade and eliciting an NK cell-mediated response against "MYC-addicted" tumors. Arrest in NK maturation has been demonstrated to be a critical mechanism of immune escape in cancers[27,28]. We demonstrate that overexpression of MYC in lymphoblasts orchestrates such arrest in NK development in the lymphoma microenvironment.

We show that MYC-driven lymphomas evade NK surveillance by transcriptionally repressing the STAT1/2-Type I IFN signaling (Fig. 5g–k, Supplementary Figs. 24–27) required for NK cell maturation[33–39]. $MYC^{High}$ lymphoblasts show suppressed Type I IFN production (Fig. 5e, f, Supplementary Figs. 20 and 22), which in turn perturbs NK maturation in the lymphoma micro-environment. We find that MYC-driven human lymphomas are associated with concomitant suppression of STAT1/2 and NK exclusion (Fig. 6).

Although we observe that MYC overexpression alters secretion of Type I IFNs from the B/T-lymphoblasts, we do not exclude the possibility that IFN secretion from other immune cells including dendritic cells (DCs) may be perturbed. Of note, we observe reductions in IL15 and IL15R in $MYC^{ON}$ mice (Supplementary Fig. 13). IL15 is essential for NK maturation and is primarily secreted by DCs following their activation by Type I IFNs[56], suggesting that loss of Type IFN secretion by MYC-transformed lymphocytes may affect DC maturation[57], consequently reduce IL15 secretion from DCs and block NK maturation. Consistent with previous studies showing an increase in DCs numbers and a block in DC maturation in MYC-driven B-lymphomas[18], we show that DC numbers are significantly increased in $MYC^{ON}$ mice (Supplementary Fig. 4e). The consequences of phenotypic and functional changes in DCs and their effects on IL15 and NK maturation during MYC-induced lymphomagenesis remain to be explored.

Type I IFN treatment restored NK cell production in spleens of lymphoma-bearing mice (Fig. 4e). IFN treatment combined with NK depletion shortened time to morbidity of $MYC^{ON}$ mice (Fig. 4g). Hence, MYC suppresses the STAT1/2-Type I IFN cascade and blocks NK surveillance (Supplementary Fig. 30). In addition to their use as antiviral agents, Type I IFNs elicit anti-proliferative effects on cancer cells[37,58]. Our study explains why MYC-driven B-lymphoma lines are sensitive to Type I IFN therapy[59], and illustrates new functional roles for Type I IFN therapy in lymphomas.

We observed that blocking IFNAR1 at the time of MYC inactivation compromised the production of CD27+CD11b+ NK effectors without suppressing the replenishment of mNK cells in the spleen (Fig. 4h, Supplementary Fig. 16). Restoration of homeostatic levels of mNK cells in $MYC^{OFF}$ mice despite IFNAR1 blockade might be explained by the activation of pathways independent of Type I IFN signaling that can induce NK maturation upon MYC inactivation.

Prior studies have shown that specific NK receptor–ligand interactions mediate the recognition of tumors by NK cells and can block MYC-driven lymphomagenesis[60,61]. Transcript levels of NKG2D (KLRK1) were reduced in $MYC^{ON}$ lymphoma mice and restored to normal levels in $MYC^{OFF}$ mice (Supplementary Fig. 13), suggesting suppression of NKG2DL on MYC-driven lymphoblasts. Therefore, multiple mechanisms of MYC-mediated suppression of NK surveillance may be at play during lymphomagenesis.

Our results have important implications for cancer immunotherapy[62]. We observed that adoptively-transferred NK cells significantly delayed MYC-driven lymphomagenesis, and contributed to sustained lymphoma regression post MYC inactivation (Fig. 7), suggesting that MYC-driven lymphomas may be particularly sensitive to NK cell-based therapies. Most conventional chemotherapies used for lymphoma are immunosuppressive. Therapies that are directed at restoring the immune response may be particularly effective against MYC-driven lymphoma. Our findings underscore the importance of identifying MYC inhibitors[63] and combining these with NK cell-based therapies[64–66] to achieve maximum therapeutic benefit. We suggest that, as others and we investigate potential therapeutic candidates that target MYC, it will be essential to consider the role of NK cell-mediated immune surveillance in defining the modes of therapy; and to identify the characteristics, timing and dosage of NK cells for maximum therapeutic benefit.

## Methods

**Cell lines and cell culture.** Conditional MYC-driven mouse T cell lymphoma cell lines (Supplementary Fig. 21) were derived from SRα-tTA/tet-O-MYC mice. c-MYC was inhibited in SRα-tTA/tet-O-MYC T-lymphoma cells by treating cell cultures with 0.02 μg/ml doxycycline (Sigma-Aldrich, T7660) for 4, 8, 24 and 48 h. BL-like human P493-6 cells were kindly provided by Chi Van Dang, University of Pennsylvania. For c-MYC inactivation in P493-6 cells, the conditional pmyc-tet construct was repressed with 0.1 μg/ml doxycycline (Sigma-Aldrich, T7660) for 24 h. Cell lines were confirmed to be negative for mycoplasma contamination and maintained in Roswell Park Memorial Institute 1640 medium (RPMI, Invitrogen) with GlutaMAX containing 10% fetal bovine serum, 100 IU/ml penicillin, 100 μg/ml streptomycin and 50 μM 2-mercaptoethanol at 37 °C in a humidified incubator with 5% $CO_2$.

**Animals.** All procedures were approved by the Administrative Panel on Laboratory Animal Care (APLAC) at Stanford University, and were carried out in accordance with institutional and national guidelines. The following strains of mice were used in the study: FVB/N (males, 8–12 weeks), SRα-tTA/Tet-O-MYC mice (males, 8–12 weeks), NOD SCID (males, 4–8 weeks), NOD SCID IL2Rγ−/− (males, 4–8 weeks), FVB/N (males, 4–8 weeks) and C57BL6J (males, 4–8 weeks). Animals were raised and housed at Stanford University and maintained at 18–23 °C, 40–60% humidity and at 12-h light/12-h dark cycle.

**Isolation and processing of immune cells from mice.** We used young male mice (8–12 weeks of age). The following strains of mice were used: SRα-tTA/tet-O-MYC mice and the corresponding FVB/N wildtype strain. The SRα-tTA/tet-O-MYC males developed T cell lymphoma at ~2–3 months of age. Diseased mice were identified based on the presence of visible symptoms of T cell lymphoma development; namely, loss of weight and appetite, ruffled fur, shortness of breath and movement difficulties and killed within 24 h of development of one or more of these symptoms. Half the number of male littermates that developed T cell lymphoma were subjected to doxycycline treatment to inactivate MYC. For T cell lymphoma mice subject to MYC inactivation, one starter dose of doxycycline was given intraperitoneally, after which, the mice continuously received doxycycline for 4 days in their drinking water. Leukemic littermates from each cage were randomly and equally divided between $MYC^{ON/HIGH}$ (untreated) or $MYC^{OFF/LOW}$ (doxycycline-treated) groups. Spleens, blood and bone marrow were isolated from the three mice groups, namely, SRα-tTA/tet-O-MYC ($MYC^{ON/HIGH}$), SRα-tTA/tet-O-MYC + Doxycycline ($MYC^{OFF/LOW}$) and normal healthy (FVB/N) controls. Blood was drawn by cardiac puncture after euthanizing the mice, and directly subjected to erythrocyte lysis. Bone marrow was obtained by either flushing the cavities of femur and tibia with PBS or by centrifugation of the bones at 1600 × g for 3 min after they have been cut in both ends. Spleen and bone marrow were homogenized using a 19$^{1/2}$ G needle and filtered through a 70 μm filter. After depletion of erythrocytes from spleen, blood and bone marrow using RBC lysis buffer (BD PharmLyse, BD Biosciences) or ACK lysing buffer (Gibco), washed cells were cryopreserved and utilized for CyTOF or flow cytometry. For processing of liver and lung cells, the mice were perfused with PBS through first the right ventricle to perfuse the lung followed by the left ventricle and the vena cava inferior to perfuse

the liver. Dissected liver and lung were cut into small pieces and digested in HBSS with $Ca^{++}$ and $Mg^{++}$ (Corning) containing 2 mg/ml Collagenase IV (Worthington) and 0.2 mg/ml DNase I (Roche) using the gentle MACS Octo Dissociator (MACS Miltenyi). Cell suspension was filtered through a 70-µm cell strainer and treated with ACK lysis buffer (ThermoFisher). The cells were utilized for flow cytometry immediately after processing. No blinding was done while executing the animal experiments.

**Mass cytometry (CyTOF Immunophenotyping)**. This assay was performed in the Human Immune Monitoring Center at Stanford University. Mouse splenocytes were thawed in warm media, washed twice, resuspended in CyFACS buffer (PBS supplemented with 2% BSA, 2 mM EDTA and 0.1% sodium azide). Viable cells were then counted by Vicell. Cells were added to a V-bottom microtiter plate at 1 million viable cells per well and washed once by pelleting and resuspension in fresh CyFACS buffer. The cells were stained for 60 min on ice with 50 µl of the following antibody-polymer conjugate cocktail (Supplementary Table 1). All antibodies were custom-ordered metal-conjugates from Fluidigm (Supplementary Table 1). The cells were washed twice; by pelleting and resuspension with 500 µl FACS buffer. The cells were resuspended in 100 µl PBS buffer containing 1.7 µg/ml Live-Dead (DOTA-maleimide (Macrocyclics) containing natural-abundance indium). The cells were washed twice by pelleting and resuspension with 500 µl CyFACS buffer. The cells were resuspended in 100 µl 2% PFA in PBS and placed at 4 °C overnight. The next day, the cells were pelleted and washed by resuspension in fresh PBS. The cells were resuspended in 100 µl eBiosciences permeabilization buffer (1× in PBS) and placed on ice for 45 min before washing twice with 500 µl CyFACS buffer. The cells were resuspended in 100 µl iridium-containing DNA intercalator (1:2000 dilution in PBS; DVS Sciences) and incubated at room temperature for 20 min. The cells were washed once in CyFACS buffer, then three times in 500 µl Milli-Q water. The cells were diluted to a concentration of ~800,000 cells/ml in Milli-Q water containing 0.1× EQ normalization beads (Fluidigm) before injection into the CyTOF (Fluidigm). Data analysis was performed using FlowJo v10 by gating on intact cells based on the Iridium vs Ce140 (bead), then both Iridium isotopes from the intercalator, then on singlets by Ir191 vs cell length, then on live cells (Indium-LiveDead minus population), followed by cell subset-specific gating (Supplementary Fig. 31). The data was analyzed using FlowJo X 10.0.7r2 and viSNE (Cytobank Inc.).

**Phospho-CyTOF**. The harvested cells were suspended in 16% PFA and incubated for 10 min at room temperature. After washing with CyFACS buffer (1× CyPBS with 0.1% BSA, and 0.05% Na Azide; made in Milli-Q water), surface staining was performed by incubating the cells with the surface-staining antibody cocktail (Supplementary Table 2) at room temperature for 30 min. The cells were then washed with CyFACS buffer and permeabilized before carrying out intracellular staining. The cells were then incubated with the intracellular antibody cocktail (Supplementary Table 2) at room temperature for 30 min. After washing with CyPBS, the cells were incubated with Ir-intercalator solution on ice for 20 min. The samples were then washed with Milli-Q water and acquired on the CyTOF instrument using standard instrument set up procedures. The data were analyzed using FlowJo X 10.0.7r2.

**Flow cytometry**. All antibodies for flow cytometry measurements are indicated in Supplementary Table 3. All antibodies were titrated prior to use to determine the dilution to use for optimal staining index. Multicolor flow cytometry staining was performed according to standard procedures for surface staining and the cells were blocked with Rat anti-mouseCD16/32 Fc block (BD Pharmingen). Cells were stained with propidium iodide (PI, Sigma-Aldrich) or LIVE/DEAD Fixable Near-IR Dead Cell Stain kit (ThermoFisher) to gate out dead cells. Labeled cells were analyzed on a FACSCalibur or a 5-laser LSRII (Becton Dickinson). Compensation was done using Ultracomp eBeads Compensation Beads (Invitrogen) and ArC Amine Reactive Compensation Bead Kit (ThermoFisher). The data were analyzed using FlowJo X 10.0.7r2. Gating strategy is provided in Supplementary Fig. 31.

**Calculation of absolute immune cell counts**. Viable cells were counted using a Vicell before subjecting the single cell suspensions of spleen, blood and bone marrow to CyTOF or flow cytometry, respectively. Using the percentages of the immune populations calculated by CyTOF and flow cytometry on the same day, absolute viable cell numbers of the different immune subsets were computed.

**RNA sequencing**. A part of the splenic single cell suspensions from normal (*FVB/N*, $n = 3$), *SRα-tTA MYC$^{ON}$* ($n = 3$) and *SRα-tTA MYC$^{OFF}$* ($n = 3$) mice subjected to CyTOF were used to generate RNA. RNA sequencing was performed at the Beijing Genomics Institute (BGI) using the BGISEQ-500 instrument. The experimental pipeline is described below.

For library preparation, oligo (dT) magnetic beads were used to select mRNA with polyA tail, or hybridize the rRNA with DNA probe and digest the DNA/RNA hybrid strand, followed by DNase I reaction to remove DNA probe. After obtaining the target RNA, it was fragmented and reverse transcribed to double-strand cDNA (dscDNA) by N6 random primer. The dscDNA was end repaired with phosphate at 5′ end and stickiness 'A' at 3′ end. The adaptor with stickiness 'T' at 3′ end was ligated to the dscDNA. Two specific primers were used to amplify the ligation product. The PCR product was then denatured and the single strand DNA was cyclized by splint oligo and DNA ligase. Sequencing was then performed on the prepared library.

Primary sequencing data (raw reads) were subjected to quality control (QC). After QC, raw reads were filtered into clean reads before alignment to the reference sequences. Once alignment passed QC, we proceeded with downstream analysis including gene expression and deep analysis based on gene expression (PCA/correlation/screening differentially expressed genes). Data were filtered reads with adaptors, reads with >10% unknown bases and low-quality reads. Bowtie 2 was used to map clean reads to reference gene, and HISAT was used to map reads to reference genome (mm8). RSEM was used to compute maximum likelihood abundance estimates using the expectation maximization (EM) algorithm. FPKM method was used to calculate the expression levels of individual genes. RNA-sequencing data are provided in Gene Expression Omnibus (GSE106078).

**CIBERSORT**. Human CIBERSORT was carried out at https://cibersort.stanford.edu/ using LM22 reference matrix of gene expression in human immune subsets (19). For CIBERSORT of splenic immune compositions in normal (*FVB/N*), *SRα-tTA MYC$^{ON}$* and *SRα-tTA MYC$^{OFF}$* mice, a reference matrix was generated using ImmGen (https://www.immgen.org/, GSE15907). CIBERSORT was carried out on CyTOF-matched normal ($n = 3$), *SRα-tTA MYC$^{ON}$* ($n = 3$) and *SRα-tTA MYC$^{OFF}$* ($n = 3$) samples.

**Transplantation of T-lymphoma and bioluminescence imaging**. A MYC-addicted T cell lymphoma cell line derived from a male *SRα-tTA/tet-O-MYC* mouse (Supplementary Fig. 21) was labeled with luciferase. $3 × 10^6$ luciferase-labeled T cell lymphoma cells were then injected intravenously into two groups of immune-deficient hosts: NOD SCID (The Jackson Laboratory) and NOD-SCIDIL-2Rγ$^{-/-}$ (NSG). Male transplant recipients used were between 4 and 5 weeks of age. Leukemic engraftment and progression were tracked by bioluminescence imaging (BLI) of the transplant recipients twice every week using an in vivo IVIS 100 bioluminescence/optical imaging system (Xenogen). D-Luciferin (Promega) dissolved in PBS was injected intraperitoneally at a dose of 2.5 mg per mouse 15 min before measuring the luminescence signal. General anesthesia was induced with 3% isoflurane and continued during the procedure with 2% isoflurane introduced through a nose cone. No blinding was done while executing the animal experiments. Bioluminescence imaging data were analyzed using Living Image software (Perkin Elmer). All mouse experiments were approved by the Administrative Panel on Laboratory Animal Care (APLAC) at Stanford University, and were carried out in accordance with institutional and national guidelines.

**Adoptive transfer of NK cells into T-lymphoma-bearing mice**. $3 × 10^6$ luciferase-labeled T cell lymphoma cells were injected intravenously into 4–8-week-old NSG mice ($n = 19$). Post verification of T cell lymphoma engraftment by BLI, all mice were treated with doxycycline to inactivate MYC. 10 of the 19 mice were subjected to adoptive transfer of NK cells isolated from spleens of 4–8-week-old FVB/N mice by magnetic-activated cell sorting (MACS, NK Cell Isolation Kit II, Miltenyi Biotech). Flow cytometry was performed to confirm the purity of the isolated NK cells. $1 × 10^6$ MACS purified NK cells were injected per mouse into NSG transplant recipients bearing regressed T cell lymphoma (on doxycycline, *MYC$^{OFF}$*).

**Type I IFN treatment, NK depletion and IFNAR1 blockade**. 20,000U of Type I IFNα (R&D systems) was administered intraperitoneally to *SRα-tTA/tet-O-MYC* mice bearing overt lymphoma for 3 days. On day 3, spleens from treated mice were collected, and subjected to flow cytometry for NK and T cell analysis. PBS-treated mice were used as controls. Absolute cell counts were obtained as described in section 'Calculation of absolute immune cell counts'. For long-term administration of Type I IFNα, the mice were treated every third day with 20,000 U IFNα or PBS intraperitoneally. 100 µg anti-NK1.1 (clone PK136, BioXCell) or IgG2a control (Clone C1.18.4, BioXCell) were administered intraperitoneally every sixth day to deplete NK cells. 2.5 mg anti-IFNAR1 (clone MAR1-6A3, Leinco) or IgG1 control (clone HKSP, Leinco) were administered intraperitoneally to block IFNAR1 in *SRα-tTA/tet-O-MYC* mice bearing overt lymphoma at the same time as the mice were subjected to MYC inactivation with doxycycline treatment. After 4 days, spleen was collected and subjected to flow cytometry for NK and T cell analysis.

**Immunoblotting**. Cells were lysed in CelLytic buffer (Sigma) supplemented with 1% protease inhibitor 'cocktail' (Pierce). Protein samples were subsequently separated by electrophoresis through NuPAGE (Invitrogen) 4–12% Bis-Tris gradient gels or 4–12% pre-cast XT-Criterion gradient gels (Biorad, Hercules, CA), and were then transferred to PVDF membranes (Immobilion; Millipore). Afterwards, blots were blocked in 5% milk in TBST and incubated overnight at 4 °C with primary antibodies (Supplementary Table 4). After incubation with primary antibodies, blots were washed with 0.1% TBST three times for 10 minutes each, followed by incubation with secondary antibodies (goat anti-mouse or goat anti-rabbit antibodies, ThermoFisher Scientific) conjugated with alkaline phosphatase (AP) in 5% milk in TBST at room temperature for an hour. Washing was repeated

as above, and blots were visualized using Novex™ AP Chemiluminescent Substrate (CDP-Star™) (ThermoFisher Scientific). Blots were developed either on films or on Biorad Chemodoc MP Imaging system with Image Lab 5.1 software.

**Retrovirus production and transduction**. Transfection of retroviral constructs (Supplementary Table 5) was performed using Lipofectamine 2000 (Invitrogen) with Opti-MEM media (Invitrogen). Retroviral supernatants for infection of cells were generated after co-transfection of Phoenix cells containing the plasmids pHIT60 (gag-pol) and pHIT123 (ecotropic envelope) (provided by G.P. Nolan), with the retroviral constructs. Cells were cultured in high-glucose DMEM (Invitrogen) with GlutaMAX containing 10% FBS, 100 IU/ml penicillin, 100 µg/ml streptomycin, 25 mM HEPES, pH 7.2, 1 mM sodium pyruvate and 0.1 mM non-essential amino acids. Serum-free medium was replaced after 16 h with growth medium containing 10 mM sodium butyrate. After 8 h of incubation, the medium was changed back to growth medium without sodium butyrate. Twenty-four hours later, we harvested the viral supernatants, passed them through a 0.45-µm filter and centrifuged them twice at $2000 \times g$ for 90 min at 32 °C on 50 µg/ml RetroNectin-coated non-tissue culture-treated six-well plates. Pre-B cells ($2 \times 10^6$–$3 \times 10^6$ cells per well) were transduced by centrifugation at $600 \times g$ for 30 min and were maintained overnight at 37 °C with 5% $CO_2$ before transfer into culture flasks. All retroviral work was carried out in a BSL2 laboratory with a designated hood for virus production and transduction.

**Lentivirus production and transduction**. Lentiviral (Supplementary Table 5) transfections were performed using Lipofectamine 2000 (Invitrogen) with Opti-MEM media (Invitrogen). We produced lentiviral supernatants to infect murine cells by co-transfecting HEK 293FT cells with the plasmids psPAX2 (gag-pol; Didier Trono, Addgene 12260) and pMD2.G (VSV-G envelope; Didier Trono, Addgene 12259). Cells were cultivated in high-glucose DMEM (Invitrogen) with GlutaMAX as described above. Culture supernatants were concentrated with Lenti-X™ Concentrator (Clontech) according to the manufacturer's recommendation. The concentrated viruses were incubated with the cells for 50 µg/ml RetroNectin-coated non- tissue culture-treated 6−well plates. All lentiviral work was carried out in a BSL2 laboratory with a designated hood for virus production and transduction.

**Quantitative RT-PCR**. Quantitative real-time PCR was carried out with the SYBRGreenER mix from Invitrogen according to standard PCR conditions and an ABI7900HT real-time PCR system (Applied Biosystems). Primers for quantitative RT-PCR are in Supplementary Table 6.

**Immunohistochemistry**. Spleens and thymuses obtained from experimental mice were immersed in 20 ml of formalin (VWR) for 24 h and transferred to PBS. Paraffin embedding was carried out using standard procedures on a Tissue-TEK VIP processor (Miles Scientific), and 4 µm sections were mounted on Apex superior adhesive slides (Leica Microsystems) and stained on a Ventana Bench-Mark automated IHC stainer. Immunohistochemistry sections were mounted with mounting medium, antifade reagent (Pro-Long Gold; Invitrogen) was applied, and coverslips were sealed before acquisition of images at 25 °C on a Zeiss Axiovert 200 M inverted confocal microscope with a 40 Plan Neofluor objective using IP Lab 4.0 software (Scanalytics).

**ChIP-sequencing analysis**. Raw ChIP-sequencing files were downloaded from Gene Expression Omnibus (GEO, GSE44672, GSE36354, Supplementary Fig. 22). The FASTQ Groomer in Galaxy Software (https://usegalaxy.org/) was used to generate files that were mapped to the RefSeq database for mouse genes (mm9). The resulting bigWig files processed from Galaxy were visualized in Integrated Genome Viewer (IGV_2.3.72).

**Statistics and reproducibility**. CyTOF, flow cytometry and CIBERSORT results are shown as box plots and comparisons between any two groups were made using two-tailed Mann–Whitney $U$ test (not assuming normal distributions) with Graphpad Prism software. For all box plots, the center is at the median, minima and maxima are indicated by the whiskers, and the box represents data between 25th and 75th percentile. All remaining data are presented as mean ± s.d. The comparisons for the mean values between two sample groups were made using the two-tailed Student's $t$-test after calculating variances for each group with Graphpad Prism software. For experiments involving transplantation of cells into mice, the minimal number of mice in each group was calculated using the 'cpower' function in the R/Hmisc package (http://www.r-project.org). The Kaplan–Meier method was used for estimation of overall survival and relapse-free survival. For all survival analyses, the log-rank test was used to compare groups of transplanted mice or human patients. The R package 'survival' version 2.35–8 was used for the survival analysis. All $p$-values are two-tailed. The level of significance was set at $p < 0.05$. $p$-values: ns = non- significant; all significant $p$-values are indicated in the manuscript.

For experiments involving primary T-lymphoma mice, reproducibility of the results was ensured by drawing mice from at least three independent breeding pairs

for each experiment. Successful replication of experiments was ensured by using multiple methodologies to ensure that the same result was obtained. For example, CyTOF, CIBERSORT and flow cytometry were performed on $FVB/N$, $MYC^{ON}$ and $MYC^{OFF}$ mice to ensure that changes in NK cell homeostasis are reproducible irrespective of the methodology used. For each of these experiments, we used both matched and independent samples to ensure reproducibility. All in vitro experiments were conducted with at least two different biological replicates, drawn from independent batches/passages of cells. For qPCR studies, each biological replicate was run as three technical replicates to account for handling errors. For immunoblots, a representative of two biological replicates is shown. To further ensure rigor, cross comparison of the results was conducted between two different experimental methods that provide the same readout. For example, phospho-CyTOF changes were validated by immunoblotting. qPCR results for some experiments were further validated using RNA sequencing or publicly available microarray data from GEO. For in vivo and in vitro studies, all attempts at replication were successful. Meta-analyses of publicly available data were conducted in both T-lymphoma and B-lymphoma to further strengthen our studies showing that NK cells have an anti-tumorigenic role in both B- and T-lymphomas.

**Institutional approval**. All mouse experiments were approved by the Administrative Panel on Laboratory Animal Care (APLAC) at Stanford University, and were carried out in accordance with institutional and national guidelines.

## Data availability

We employed previously published data sets available in GEO to conduct CIBERSORT of human B- and T-ALL patients as described in Fig. 6. The following data sets were used: GSE4475 (Fig. 6a–e), GSE62156 (Fig. 6f–h) and GSE11057/GSE11058 (Fig. 6f–h). RNA-sequencing data of bulk splenic samples from normal, $SR\alpha$-tTA $MYC^{ON}$ and $SR\alpha$-tTA $MYC^{OFF}$ mice generated by us (Fig. 4a–d) can be accessed in GEO (GSE106078). All other relevant data are available in the Article, Supplementary Information or from the corresponding author upon reasonable request.

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

## Acknowledgements

We thank M. D. Leipold, R. Fernandez and Y. Rosenberg-Hasson at the Human Immune Monitoring Center, Stanford University for Luminex, CyTOF and Phospho-CyTOF acquisition. We also thank P. Chu in the Department of Pathology at Stanford University for carrying out tissue sectioning; H. Dai for technical help; A. Gentles for providing us access to PRECOG data sets; and M. Eilers for gifting us lentiviral overexpression plasmids for the study. We thank M. Eilers for his expert scientific guidance and suggestions. This work was supported by the following grants: NIH/NCI CA170378PQ2 (D.W.F.), 1U01CA188383-01/S1 (D.W.F.), Emerson Collective Cancer Research Fund (D.W.F.), the Special Fellow Award LLS 3366-17 from The Leukemia and Lymphoma Society (S.S.) and American Society of Hematology Scholar Award (S.S.). A.S.H. was funded by Grant 8026-00018B from Independent Research Fund Denmark, L.D.H. was funded by the Lundbeck Foundation, Denmark, R.D was funded by K08 National Cancer Institute (CA222676), and D.F.L. was supported by Tumor Biology Training Grant (NIH 5T32CA009151-38), Stanford University (National Cancer Institute), Burroughs Wellcome Fund Postdoctoral Enrichment Award and Research Supplement Award (National Cancer Institute).

## Author contributions

S.S. and D.W.F. conceived the study. S.S. and A.S.H. developed experimental methodology, conducted experiments and interpreted results. L.D.H., A.D., W.D.M.F., D.F.L., C.H., A.M. and M.L. performed experiments. R.D. performed analysis and interpretation of RNA-sequencing results. H.T.M. provided helpful scientific input on the manuscript. S.S., A.S.H. and D.W.F. wrote the manuscript. S.S., H.T.M. and D.W.F. provided administrative, technical and material support. D.W.F. supervised the study.

## Competing interests

The authors declare no competing interests.
