## [Peer Review File · Nature Communications]

Reviewers' comments:

Reviewer #1 (Remarks to the Author):

This is a novel, carefully controlled study in SR α -tTA MYC-ON/OFF transgenic mice rigorously demonstrating, for the first time, that activation of an oncogene (MYC) during malignant transformation (T cell lymphomagenesis) adversely impacts NK cell maturation. Further, the authors show this inhibition of NK maturation is the result of decreased STAT1/2 activation, which in turn promotes tumor progression (and can be reversed via activation of STAT1/2 with Type I IFNs). Striking clinical correlations to lymphoma patients with activation of MYC are also shown. A very nice and complete study from a strong research team led by Dr. Felsher that will be of interest to the readership as the subject of novel mechanisms of innate immune evasion is of high interest to both the cancer and the immunology readership. I have only a few minor comments:

1) CD46(+) NK T cells have been described (Yu et al J Clin Invest. 2011 Apr;121(4):1456-70. doi: 10.1172/JCI43242. PMID: 21364281), and they appear to diminish with MYCON and return in MYCOFF in Suppl Fig 5. If possible, please estimate if this decrease (percent and/or absolute number) was significant as well.

2) Long-term administration of Type I IFNs to SR α -tTA MYCON mice bearing overt T cell lymphomas (n = 8) improves overall survival (OS) in comparison to vehicle (PBS)-treated (n = 8) lymphoma-bearing mice (Figure 4k). It would be interesting to know if a similar experiment was performed following NK depletion, and if performed, whether or not OS advantage associated with IFN- α administration was lost. This would provide additional evidence to the evidence/conclusions by the authors that IFN- α is acting directly on NK cell maturation without a significant direct anti-tumor effect.

3) It would also be interesting to know if the authors documented decreased STAT1/2 in the NKP or immature NK cell subset in SR α -tTA MYCON mice. Did this increase in SR α -tTA MYCOFF mice?

4) The assessment of STAT1/2 levels in NK cells from patients with T cell lymphomas (compared to controls) is impressive (Fig 6). Were the controls aged matched to the patients?

5) Tumor intrinsic changes in STAT1/2 signaling were found within the lymphomas in the SR α -tTA MYCON mice bearing overt T cell lymphomas or in their isolated lymphoma? Was it assumed or documented that the source of the lost IFN- α during lymphomagenesis was restricted to the SR α -tTA MYCON lymphomas? If assumed, can the authors speculate on whether these SR α -tTA MYCON mice bearing overt T cell lymphomas were likely to cause a decrease IFN- α from sources other than the tumor cells themselves?

6) in general, the statistics look robust. I have only one concern that the ratio of total NK among PBMC from normal healthy vs T ALL patients is correctly scored as significant (Fig 6).

Michael A. Caligiuri, MD

Reviewer #2 (Remarks to the Author):

Understanding how cell intrinsic oncogenic pathway in tumors can affect tumor microenvironment is a very interesting topic especially given the growing number of targeted therapies tested in cancer. The tumor mechanisms that restrict immune responses against tumor is also a hot topic especially with regards to NK cell that represent key effector immune cells against hematological malignancies and metastasis. In this study Swaminathan et al demonstrate a systemic reduction of

natural killer (NK) in mice bearing MYC-driven T-lymphomas, due to an arrest in NK cell maturation. They show that inactivation of lymphoma-intrinsic MYC releases the brakes on NK maturation suggesting that tumor MYC expression directly affect NK cell homeostasis. They suggest that Lymphoma-intrinsic MYC arrests NK maturation by transcriptionally repressing type I Interferons (IFNs). The finding from the authors are well presented, retrospective human data were included and a molecular mechanism was exposed. For all these reasons, the study presented here is of real interest. Still several issues summarized below actually limit the conclusions from the authors. 1st although the authors focused on NK cells their functional role in their MYC on off model remains to be clearly demonstrated. 2nd the maturation of NK cells that is supposed to be the origin of MYC driven defects was not deeply analysed since no classical NK cell maturation analysis was shown on any figures. 3rd although T-lymphoma-bearing mice treated with Type I IFN have restored NK and improved survival, the role of IFN type I in MYC driven defect require additional confirmations especially given conflicting results from the literature concerning the impact of type I IFN on NK cell development.

Major points :

1. The authors suggest that MYC-driven lymphomas evade immune surveillance by suppressing NK cell. Unfortunately functional experiments showing increased tumor growth or persistence upon MYC inactivation in the absence of NK cells are missing in their original MYC driven lymphoma model. NK cell depletion at the MYC on or MYC off stage of lymphoma should be performed using anti-ASGM1 or anti-NK1.1 mAbs that are commonly used to study NK cell role in tumor mouse model. Alternatively NCR1crexDTR could be used.

2. The authors state that "NK cells were the sole subset significantly altered both in absolute cell numbers (Figure 1h) and percentages (Figure 1d, Supplementary Figures 2e-f)." Although the authors show that NK cells percentages and numbers are modulated during lymphoma development unlike B cells, DCs and neutrophils, I suggest the authors to remove "sole" as they did not show whether non malignant T cells (CD4+ Tconv, Tregs, CD8+ T cells, NKT cells, gd T cell) number or percentages are altered. In addition supplementary figures 5b and 7a suggest CD8+ T cells SP percentages are greatly reduced upon lymphoma development. I suggest the authors to show CD8+ T cells numbers and percentages evolution upon MYC activation and inactivation to determine whether this subset is also impacted by MYC. If CD8 T cells are modulated by MYC, the authors should also determine the importance of this subset relative to NK cells in antitumor responses using anti-CD8b depleting antibodies.

3. The authors state that "NK cell suppression during MYC-driven lymphomagenesis is systemic" through the analysis of NK cells in the BM, blood and spleen that may all contain drastic elevation of lymphoma T cells in MYCon tumor bearing mice. It would be interesting to determine in other NK cell compartments such as liver and lung that may contain less tumors whether similar alteration in NK cell numbers/development occurs.

4. The authors demonstrated the dependency of NK suppression on MYC status, by comparing percentages NK cells in MYC OFF mice with residual blasts alongside lymphoma-bearing mice MYC ON with similar blast percentages. addressing whether NK cell suppression is the consequence of lymphoma burden or MYC expression/absence is a key point of this study. To strengthen their result I would suggest the authors to add correlations between NK cell percentages/numbers (or NKp46 MFI) and lymphoma percentages/numbers in MYCOFF and MYCON tumors. We would expect NK cells to be inversely correlated with MYCON but not MYCOFF tumors. With regards to MYCOFF tumors persistence, is there any correlation between NK cell percentages and tumor persistence? Is the tumor persistence affected by NK cell absence, experiments using depleting antibodies (e.g anti-AsGM1 mAbs) could be performed. This is a particularly important point as the authors imply that NK cell alteration upon lymphoma development impact lymphoma development.

5. The authors found that mice bearing overt MYC-driven lymphomas have reduced number of NK cells because of a developmental blockade at the NK precursor (NKP) stage that prevents the

production of immature NKp46+ iNK cells. They compared the absolute counts of NK cell precursors (NKP, CD122+ NKp46-, Figure 3a), and the more differentiated CD122+ NKp46+ iNK (immature NK, Figure 3b) cells in bone marrow from normal (n = 8), SR α -tTA MYCON (lymphoma, n = 8), and SR α -tTA MYCOFF (regressed lymphoma, n = 8) mice. Given the absence of proper markers assessing NK cell maturation (CD11b, CD27, KLRG1; Huntington ND et al, *J Immunol.* 2007 Apr 15;178(8):4764-70.; Chiossone L, et al, *Blood.* 2009 May 28;113(22):5488-96. doi: 10.1182) I would strongly recommend to avoid the term of iNK as the CD112+NKp46+ NK cells, even in the bone marrow, contain true iNK (CD27+CD11b-, Ly49-, NKG2A+) but also mature NKs (CD11b+CD27+ and CD11b+CD27-). Given the functional differences between iNK and mNK, it is critical to understand whether NK cell maturation stages are differentially affected by MYC. A CD11b CD27 assessment of NK cells in the BM, spleen,(liver and lung are also interesting as they have respectively more CD27SP or CD11B SP) of MYCon and MYCoff tumors should be done. In addition although the authors focused mainly on NKp46 the expression of other key NK cell activating (NKG2D, DNAM-1 for example; Martinet L et al *Nat Rev Immunol.* 2015 Apr;15(4):243-54. doi: 10.1038/nri3799.) or inhibitory receptors (NKG2A, Ly49s) should be performed to determine more extensively the consequence of MYC driven NK cell alteration. Again correlation of NK cell maturation stages and tumor burden in MYCON and OFF should be shown to understand the relative importance of MYC expression/tumor burden on NK cell development. In addition functional analysis of NK cells upon MYC tumor development may be decisive to demonstrate MYC driven NK cell dysfunctions.

6. The authors show that MYC-driven lymphomas evade NK cell-mediated immune surveillance by transcriptionally repressing Type I IFN cytokine axis required for NK cell maturation. However, previous studies including those cited by the authors do not show a critical impact of IFN type I signaling on NK cell frequency and most of them only revealed minor developmental alterations. IFNAR signaling did not affect NK cell number or frequency and led to slightly iNK decrease and mNK increase according to Guan, J. et al (*PLOS ONE* 9, e111302 2014). Mizutani T et al (*Oncoimmunology.* 2012 Oct 1;1(7):1027-1037.) showed no significant differences in NKP, NKi and NKm percentages in the bone marrow of *Ifnar1*^{-/-} mice. In the spleen they found increased iNK CD27+ CD11b- and decreased mNK CD27-CD11b+ suggesting that IFNAR deficiency may block NK cell final maturation stage rather than blocking the transition from NKP to NKi. Swann et al (*J Immunol.* 2007 Jun 15;178(12):7540-9.) did not find striking alteration in the expression of the maturation markers (CD122, CD11b, and Ly49 CI). These studies using IFNAR KO mice questions whether IFN-I alteration upon MYC activation could be the main mechanism leading to the strong NK cell decrease. To confirm their hypothesis the authors should perform experiments using anti-IFNAR mAbs to determine whether IFNAR blockade prevent NK cell replenishment upon MYC inactivation in their MYCon MYCoff model.

7. In addition in the same study Mizutani showed NK cells maturation defects in *Ifnar1*^{-/-} mice and not in *Ifnar1*^{f/f} *Ncr1*-iCre mice. This suggests that Type I IFN is essential for NK cell maturation but does not require the presence of the Type I IFN receptor on the NK cells themselves, a point that should be at least discussed in the present manuscript since it would suggest that MYC blockade of IFN α signaling affect NK cells indirectly.

8. The authors show that malignant lymphocytes overexpressing oncogenic MYC show suppressed production of Type I IFNs (Figures 5e-f, Supplementary Figures 10, 15). they conclude that silenced type I signaling by lymphoma cells leads to NK cells maturation blockade. Although data showing that MYC can directly affect type I IFN are convincing it is difficult to understand why tumor cell derived type I IFN is required for NK cell maturation since NK cell maturation occurs in the absence of lymphoma in naive mice. The authors should investigate whether lymphoma expressing *myc* can also affect type I IFN production by cells from the microenvironment (DC for example). It is reasonable to speculate that type I IFN silencing by MYC could limit tumor immunogenicity leading to poor NK cell priming and killing but it is difficult to understand why tumor derived IFN I is required for NK cell development especially if type I IFN are normally produced by other cells form the microenvironment . The authors should at least discuss

alternative hypothesis. For instance IL-15 was previously shown to be critical for NK cell development and priming (Huntington ND, J Exp Med. 2009 Jan 16;206(1):25-34.) and IL-15 was shown to be induced by type I IFN on DCs (Ganal SC, Immunity. 2012 Jul 27;37(1):171-86. doi: 10.1016/j.immuni.2012.05.020). IL-15 shortage upon lymphoma development should be envisaged.

Reviewer #3 (Remarks to the Author):

Swaminathan et al., report that MYC drive T cell lymphomas drive developmental defects in splenic B and NK cells. Inducible repression of MYC rescues NK and B cell development. The authors propose a mechanism by which MYC-driven lymphomas decrease type I interferon production to block NK cell development and subsequent tumor surveillance. Cytokine treatment with IFN- α is sufficient to restore NK cell numbers and increase survival in MYC ON mice. However, there are quite a few key issues with the proposed hypothesis, with additional detail needed to resolve these issues:

Major Issues

1. The authors do not correctly identify iNK, NKP, and mature NK cell populations in the spleen and bone marrow. NKP = lin⁻ (CD19, CD3, TCR α) CD122+NKp46-NK1.1⁻, iNK = lin⁻ CD122+NKp46+NK1.1+CD49b⁻, mNK = lin⁻CD122+NKp46+NK1.1+CD49b⁺. These errors are apparent in Fig 3H with a drastic increase in the proposed "NKP" population, which would likely be gated out using lineage markers. The data in figure 3 are insufficient to correctly identify these populations and need to be redone.

2. The authors claim that NK cell development is perturbed and need to strengthen these data by including other markers commonly used to measure NK cell maturation in addition to NKp46 (CD11b, CD27, KLRG1).

3. The authors incorrectly claim that previous studies show a direct role for type I interferon in NK cell development. These studies show that NK cell frequencies and numbers are largely similar between WT and IFNAR-deficient mice, and that there is no cell-intrinsic role of type I interferons in NK cell development (Madera et al., 2016 JEM; Guan et al., 2014 Plos One; Baranek et al., 2012 Cell Host Microbe) These data contrast to the developmental defects observed in the MYC ON mice. Thus, the possibility remains that in vivo IFN- α treatment has cell-extrinsic effects on NK cell development and or proliferation in the MYC ON state, possibly by driving increased IL-15 production in vivo. Because both IL-15 and IL-15 α expression are increased in the spleens of WT and MYC OFF mice, and DC homeostasis is not perturbed in the MYC ON state, the author's need to formally address or disprove the possibility that decreased systemic levels of IL-15 can explain their defects in NK development.

4. The authors claim that MYC driven T cell lymphomas suppress type I interferon production to perturb NK cell development. The authors need to demonstrate that deletion of type I interferons, specifically in MYC OFF lymphomas, is required for the observed rescue in NK cell development.

Minor

1. The authors need to demonstrate that there is no effect of doxycycline treatment on NK cell development (in WT mice). This is important because previous studies have shown that acute antibiotic treatment can impact NK cell homeostasis (Kamimura et al., 2015 Cell Reports).

2. The authors should look in the liver and lung of MYC ON and MYC OFF mice to determine if the

decrease in NK cells is not confined to discrete tissue microenvironments (spleen and bone marrow).

SUMMARY OF KEY NEW FINDINGS

We thank the Reviewers for their productive and valuable suggestions that has enabled us to submit a highly improved and revised manuscript. Please note that all New Figures in the revised manuscript are highlighted in blue and grey throughout the rebuttal.

Below we summarize the key new *in vivo* and *in situ* studies we conducted as the Reviewers suggested to understand the causal mechanisms by which the MYC oncogene suppresses NK cell-mediated immune surveillance:

- (1) **Manuscript lines 223-288:** We find that oncogenic MYC blocks both early and late stages of NK cell maturation during lymphomagenesis. We observed changes in NK cells both in the tumor microenvironment *in situ* but also in other organs and tissues including: bone marrow, spleen, lung, and liver of normal (healthy), MYC-ON (overt T-cell lymphoma) and MYC-OFF (regressed T-lymphoma) mice (**Figure 3 and Supplementary figure 11**). To conduct these experiments, we have employed a new flow cytometry analyses including key markers of early and late NK maturation (**Supplementary Table 3**) as suggested by the Reviewers. Gating strategy for delineating the specific NK development stages is described in **Supplementary Figure 25**.
- (2) **Manuscript lines 223-288:** By systematically measuring NK cell maturation as described in point (1), we found that MYC suppresses NK cells in two ways:
 - First, overexpression of MYC in T-lymphoblasts causes reduction in numbers of all Lin-CD122+ NK cells by blocking very early NK cell maturation before the development of NK precursors (NKP) in the bone marrow (**Figures 3a-c- see below**). This early maturation block in the bone marrow leads to a reduction in mature NK cell frequencies in the spleen (**Figures 3f-g- see next page**).
 - Second, MYC overexpression in T-lymphoblasts of MYC-ON mice perturbs the homeostatic distributions of residual mature splenic NK effector subsets (**Figures 3h-i- see next page**), thereby likely compromising NK cell function during MYC-induced lymphomagenesis.

Figure 3: Maturation of NK cells is arrested in mice bearing overt MYC-driven T cell lymphomas

(3) **Manuscript lines 340-362:** We previously described that Type I IFN treatment is sufficient to restore NK homeostasis and prolong overall survival of overt T-cell lymphoma-bearing MYC-ON mice (**Figure 4f**). In the revised manuscript, we conducted two new functional *in vivo* experiments to establish causality between Type I IFN signalling and NK surveillance in MYC-driven lymphomas:

-First, we show that blockade of Type I IFN signalling at the time of MYC inactivation in transgenic SR α -tTA-Tet-O-MYC mice bearing overt T-lymphomas leads to reduction in frequencies of mature CD27+ CD11b+ NK effectors in comparison to mice where MYC inactivation was not combined with IFN α R1 blockade (**Figures 4h-i- see below**). Therefore, we conclude that Type I IFN signalling is required for restoring the homeostatic frequencies of functional NK effectors upon MYC inactivation.

-Second, we depleted NK cells during administration of IFN α to overt T-lymphoma bearing MYC-ON mice. We observed that there is a trend towards reduced overall survival in mice where NK cells were depleted as compared to mice that received IFN α treatment alone (**Figure 4g- see right**). Therefore, we conclude that the NK compartment is at least in part required for the anti-lymphoma effects of Type I IFN.

(4) **Manuscript lines 453-467:** By depleting NK cells before the development of overt MYC-driven T-lymphomas in SR α -tTA/Tet-O-MYC mice, we show that NK cells are required for suppressing the development of aggressive T-lymphomas that manifest with splenomegaly (**Supplementary figure 22- see below**).

Supplementary Figure 22: *Depletion of NK cells increases aggressiveness of MYC-driven T-lymphomas*

Below, we address each of the Reviewers' individual concerns. Please find highlighted the above-described **New Figures** in our responses to the Reviewers individual concerns. Please also note that **Figures not included in the final manuscript are shown in response to the Reviewers specific concerns.**

Reviewer #1:

This is a novel, carefully controlled study in SR α -tTA MYC-ON/OFF transgenic mice rigorously demonstrating, for the first time, that activation of an oncogene (MYC) during malignant transformation (T cell lymphomagenesis) adversely impacts NK cell maturation. Further, the authors show this inhibition of NK maturation is the result of decreased STAT1/2 activation, which in turn promotes tumor progression (and can be reversed via activation of STAT1/2 with Type I IFNs). Striking clinical correlations to lymphoma patients with activation of MYC are also shown. A very nice and complete study from a strong research team led by Dr. Felsher that will be of interest to the readership as the subject of novel mechanisms of innate immune evasion is of high interest to both the cancer and the immunology readership. I have only a few minor comments:

We thank the Reviewer for reviewing our manuscript, for their positive comments, and for considering our study to be a novel and carefully controlled study. To address the Reviewer's concerns, we have conducted additional experiments as well as reanalysed and discussed some of our previously existing data as described below. Please note that all **New Figures** in the revised manuscript are highlighted in blue and grey throughout the rebuttal. Please also note that Figures not included in the final manuscript are shown in response to the Reviewers specific concerns.

- (1) CD46(+) NK T cells have been described (Yu et al J Clin Invest. 2011 Apr;121(4):1456-70. doi: 10.1172/JCI43242. PMID: 21364281), and they appear to diminish with MYCON and return in MYCOFF in Suppl Fig 5. If possible, please estimate if this decrease (percent and/or absolute number) was significant as well.

We thank the Reviewer for the thoughtful suggestion of evaluating changes in the NKT compartment upon MYC activation and inactivation in our MYC-driven T-lymphoma mouse model. To address this question, we have analyzed our CyTOF data comparing normal, MYC-ON and MYC-OFF T-lymphoma mice (**Figure 1, Supplementary Figures 2 and 4**). As the Reviewer correctly points out, we found a statistically significant reduction in both percentages and absolute cell numbers of CD3+ NKp46+ NKT cells in the MYC-ON mice in comparison to normal and MYC-OFF mice (**Supplementary Figures 2f-g and Supplementary Figure 4d, Manuscript text: lines 139-140, 16-162**), suggesting that NKT cells are also compromised in MYC-driven lymphoma.

- (2) Long-term administration of Type I IFNs to SR α -tTA MYCON mice bearing overt T cell lymphomas (n = 8) improves overall survival (OS) in comparison to vehicle (PBS)-treated (n = 8) lymphoma-bearing mice (Figure 4k). It would be interesting to know if a similar experiment was performed following NK depletion, and if performed, whether or not OS advantage associated with IFN- α administration was lost. This would provide additional evidence to the evidence/conclusions by the authors that IFN- α is acting directly on NK cell maturation without a significant direct anti-tumor effect.

We appreciate the Reviewer's suggestion. As suggested by the Reviewer, we addressed whether depleting NK cells during IFN α administration decreased overall survival by treating overt T-lymphoma bearing SR α -tTA MYC ON mice with either anti NK1.1 NK cell depleting antibody (n = 3) or IgG2a control antibody (n = 3) in combination with recombinant Type I IFN α . We observed that depleting NK cells shortened overall survival times in contrast with the control group where NK cells were not depleted (**Figure 4g, Manuscript text: lines 340-347**). Hence, we conclude that anti-tumor effects of Type I IFN are at least in part dependent on NK cells. These findings further corroborate our earlier observations that Type I IFN exerts its anti-tumor effects by not directly acting on the T-lymphoblasts and rather on the host immune cells (**Figure 4e and Supplementary figure 13, Manuscript text: lines 329-339**).

- (3) It would also be interesting to know if the authors documented decreased STAT1/2 in the NKP or immature NK cell subset in SR α -tTA MYCON mice. Did this increase in SR α -tTA MYCOFF mice?

We thank the Reviewer for this thoughtful comment. We could demonstrate by RNA sequencing of bulk splenic samples in normal, MYC-ON T-lymphoma and MYC-OFF T-lymphoma mice that there is a reduction in mRNA levels of both STAT1 and STAT2 in MYC-ON mice in comparison to normal and MYC-OFF groups (**Figures 4a-d, GSE106078, Figure shown below, Manuscript text: lines 293-303**). We could not evaluate the mRNA and protein levels of STAT1 and STAT2 in specific NK subsets in the MYC-ON mice because of the technical difficulties in obtaining sufficient NK cells in the MYC-ON mice. We would like to kindly point out to the Reviewer that total NK numbers (CD3- CD122+ NKp46+) are drastically reduced in MYC-ON mice (Figures 1-3) making it a challenge to obtain sufficient numbers of NK cells from these mice for downstream analysis of STAT1-STAT2 levels. However, based on bulk RNA sequencing data comparing normal, MYC-ON and MYC-OFF mice (**Figures 4a-d, GSE 106078, Figure shown to right**), and published findings demonstrating the requirement of STAT1 downstream of Type I IFN signaling in NK cells (Liang *et al.*, Cytokine, 2003, PMID: 12967644), we speculate that STAT1/2 are reduced in specific NK subsets in MYC-ON mice compared to healthy and MYC-OFF groups.

- (4) The assessment of STAT1/2 levels in NK cells from patients with T cell lymphomas (compared to controls) is impressive (Fig 6). Were the controls aged matched to the patients?

This is a very thoughtful and important concern raised by the Reviewer. The controls in **Figure 6g-h** are not age-matched to the patients. We would like to bring to the Reviewer's attention that the patients with T-lymphoma whose blood samples have been analyzed in Figure 6 were all pediatric patients (**GSE62156**, Peirs *et al.*, Blood, 2014, PMID: 25301704; Clappier *et al.*, JEM, 2011, PMID: 21464223). Obtaining age and tissue-matched samples from healthy pediatric individuals was unfortunately prohibitively difficult. Hence, we used data from adults 18-40 years of age as our healthy counterparts for comparative analysis. We have now modified this text to discuss this pitfall of our analyses and made suggestions to address this in future experiments.

- (5) Tumor intrinsic changes in STAT1/2 signaling were found within the lymphomas in the SR α -tTA MYCON mice bearing overt T cell lymphomas or in their isolated lymphoma? Was it assumed or documented that the source of the lost IFN- α during lymphomagenesis was restricted to the SR α -tTA MYCON lymphomas? If assumed, can the authors speculate on whether these SR α -tTA MYCON mice bearing overt T cell lymphomas were likely to cause a decrease IFN- α from sources other than the tumor cells themselves?

We thank the Reviewer for raising this interesting question. The Reviewer's first question is whether

it was assumed or documented that the source of the lost IFN α was the MYC-transformed T-lymphoblasts in T-lymphoma or the B-lymphoblasts

in B-ALL. We documented that one of the sources of the lost Type I IFN were the T- and B-lymphoblasts themselves (**Figures 5d-f, Supplementary figures 17 and 19c**).

However, as the Reviewer correctly points out, we do not exclude that MYC-driven leukemogenesis disrupts the homeostasis of key Type I IFN producing immune subsets including Dendritic Cells (DCs, Montoya *et al.*, Blood, 2012, PMID: 11964292). In this regard, we do observe a reduction in IL15 and IL15R in MYC-ON mice (**Figure on previous page**) suggesting that loss of Type IFN secretion by the MYC-transformed lymphocytes may be affecting DC maturation (Montoya *et al.*, Blood, 2012, PMID: 11964292), and consequently reducing the levels of IL15 (Mattei *et al.*, JI, 2001, PMID: 11466332), a cytokine essential for NK cell maturation and function (Carson *et al.* J Exp Med, 1994, PMID: 7523571).

Consistent with previous studies showing an increase in DCs numbers and a block in DC maturation in MYC-driven B-lymphomas (Rehm *et al.*, Nat Commun, 2014, PMID: 25266931), we do show that pan CD11c⁺ DC numbers are significantly increased in MYC-ON mice (**Supplementary Figure 4e**). Respectfully, we believe that formally studying the phenotype and function of DCs and their effects on NK maturation in our inducible MYC-driven T-lymphoma model would constitute an entirely separate study. Hence, we have discussed these possibilities in the manuscript (**Manuscript lines 541-552**), and will interrogate how DCs affect NK maturation in MYC-driven lymphomas in future studies from our group.

(6) In general, the statistics look robust. I have only one concern that the ratio of total NK among PBMC from normal healthy vs T ALL patients is correctly scored as significant (Fig 6).

We thank the Reviewer for the positive comment regarding the robustness of our analyses. Per the Reviewer's suggestion, we performed statistics to compare the total NK cell percentages between healthy controls and T-lymphoma patients (**Figure 6j**) using the Mann-Whitney test, not assuming normal distributions with two-tailed p-values. We observed that the findings are significant. Also, we would like to point out that the medians between the two groups are statistically different, and the significance could be a result of a sufficiently large sample size in both groups (**Figure 6j**, line in the box plot indicates the median).

Reviewer #2:

Understanding how cell intrinsic oncogenic pathway in tumors can affect tumor microenvironment is a very interesting topic especially given the growing number of targeted therapies tested in cancer. The tumor mechanisms that restrict immune responses against tumor is also a hot topic especially with regards to NK cell that represent key effector immune cells against hematological malignancies and metastasis. In this study Swaminathan et al demonstrate a systemic reduction of natural killer (NK) in mice bearing MYC-driven T-lymphomas, due to an arrest in NK cell maturation. They show that inactivation of lymphoma-intrinsic MYC releases the brakes on NK maturation suggesting that tumor MYC expression directly affect NK cell homeostasis. They suggest that Lymphoma-intrinsic MYC arrests NK maturation by transcriptionally repressing type I Interferons (IFNs). The findings from the authors are well presented, retrospective human data were included and a molecular mechanism was exposed.

For all these reasons, the study presented here is of real interest. Still several issues summarized below actually limit the conclusions from the authors. 1st although the authors focused on NK cells their functional role in their MYC on off model remains to be clearly demonstrated. 2nd the maturation of NK cells that is supposed to be the origin of MYC driven defects was not deeply analysed since no classical NK cell maturation analysis was shown on any figures. 3rd although T-lymphoma-bearing mice treated with Type I IFN have restored NK and improved survival, the role of IFN type I in MYC driven defect require additional confirmations especially given conflicting results from the literature concerning the impact of type I IFN on NK cell development.

We thank the Reviewer for taking the time to review our manuscript, for considering our study to be interesting and topical, and for finding our data to be well-presented. The Reviewer's suggestions have enabled us to markedly improve our study, and gain a better understanding of how the MYC oncogene causally regulates NK surveillance in lymphoid malignancies. Below, we describe the additional experiments we conducted and the textual changes we made to address the Reviewer's specific concerns. Please note that all **New Figures** in the revised manuscript are highlighted in blue and grey throughout the rebuttal. Please also note that Figures not included in the final manuscript are shown in response to the Reviewers specific concerns

Major points:

- (1) The authors suggest that MYC-driven lymphomas evade immune surveillance by suppressing NK cell. Unfortunately, functional experiments showing increased tumor growth or persistence upon MYC inactivation in the absence of NK cells are missing in their original MYC driven lymphoma model. NK cell depletion at the MYC on or MYC off stage of lymphoma should be performed using anti-ASGM1 or anti-NK1.1 mAbs that are commonly used to study NK cell role in tumor mouse model. Alternatively, NCR1crexDTR could be used.

We thank the Reviewer for their comment. We agree that functional experiments demonstrating the requirement and sufficiency of NK cells in delaying lymphoma initiation and lymphoma relapse are necessary. To address this comment, we have performed the following experiments:

- Requirement of NK cells in slowing down growth of MYC-ON lymphomas:
We started depletion of NK cells using anti-NK1.1 antibody in 4-week-old SR α -tTA-MYC ON mice prior to overt lymphoma development at 8-12 weeks, and compared these to IgG2a-treated control mice (**Supplementary Figure 22**). We observed that 100% of anti NK1.1-treated mice succumbed to aggressive lymphomas accompanied by splenomegaly (**Supplementary Figures 22b-c**), while control antibody treated mice showed a heterogeneous pattern of disease aggressiveness with only 50% of the mice developing splenomegaly. Comparison of Disease Free Survival Probabilities indicated slightly prolonged survival in the control group in comparison to the NK-depleted group albeit this was not statistically significant (**Supplementary Figure 22a**). We predict that increasing the sample sizes to include more mice per group could lead to statistically significant changes in DFS between the two groups. Of note, we observed a statistically significant increase in the

survival probabilities of mice treated with the control antibody when morbidity was measured as a function of splenomegaly which indicates high disease severity (Supplementary Figure 22b). Hence, we conclude that NK cells are essential for blocking MYC-driven lymphomagenesis (Manuscript lines 453-467).

We corroborated our results by comparing the time to lymphoma development by Bioluminescence Imaging in NSG (no NK cells) and NOD-SCID (with NK cells) transplant recipients (Figures 7a-b). We observed that the presence of NK cells in NOD-SCID mice significantly prolonged the time to T-lymphoma engraftment in comparison to their NSG counterparts (Figure 7b), suggesting that NK cells are required to block the initiation of MYC-driven lymphomas (Manuscript lines 468-478).

- NK cells are sufficient for slowing down growth of MYC-ON lymphomas: We showed by adoptive transfer of syngeneic NK cells prior to lymphoma transplantation that transfer of NK cells alone is sufficient to slow down the growth of MYC-driven lymphomas (Figures 7c-d, Manuscript lines 479-490).
- NK cells are sufficient for slowing down recurrence of MYC-driven lymphomas post MYC inactivation: We showed by adoptive transfer of syngeneic NK cells at the time of MYC inactivation in T-lymphoma transplant recipients that NK cells are sufficient to block the recurrence of MYC-driven lymphomas post MYC inactivation (Figures 7e-g, Manuscript lines 491-498).

We performed a long-term survival study to compare T-lymphoma recurrence post MYC inactivation in primary SR α -tTA/Tet-O-MYC mice with and without NK depletion. This is because we have shown in previous studies that the sustained lymphoma regression and low recurrence rate post MYC inactivation (Choi *et al.*, PNAS, 2011, PMID: 21969595) in SR α -tTA-Tet-O-MYC mice is a consequence of both lymphoblast-intrinsic and extrinsic processes. Hence, we speculate that although NK depletion may accelerate lymphoma relapse, to see a significant difference, we would need to carry out long-term treatment spanning over several months with NK depleting or control antibodies combined with MYC inactivation. Respectfully, we believe that such a long-term survival study is beyond the scope of our current manuscript.

- (2) The authors state that “NK cells were the sole subset significantly altered both in absolute cell numbers (Figure 1h) and percentages (Figure 1d, Supplementary Figures 2e-f).” Although the authors show that NK cells percentages and numbers are modulated during lymphoma development unlike B cells, DCs and neutrophils, I suggest the authors to remove “sole” as they did not show whether non-malignant T cells (CD4+ Tconv, Tregs, CD8+ T cells, NKT cells, gd T cell) number or percentages are altered. In addition, supplementary figures 5b and 7a suggest CD8+ T cells SP percentages are greatly reduced upon lymphoma development. I suggest the authors to show CD8+ T cells numbers and percentages evolution upon MYC activation and inactivation to determine whether this subset is also impacted by MYC. If CD8 T cells are modulated by MYC, the authors should also determine the importance of this subset relative to NK cells in antitumor responses using anti-CD8b depleting antibodies.

We thank the Reviewer for making this thoughtful suggestion. As advised by the Reviewer, we have now modified the statement by removing the word ‘sole’ to describe the changes in NK composition during MYC-driven lymphomagenesis. This is because as the Reviewer correctly points out we did observe significant decreases in other immune subsets including $\gamma\delta$ T cells (Supplementary figures 2c and 4c) and NKT cells (Supplementary Figures 2f and 4d) in MYC-ON mice in comparison to normal and MYC-OFF groups. However, we would like to kindly point to the Reviewer that our MYC-driven lymphoma model is a T-lymphoma where disruption in a specific T cell subset leads to concomitant decreases in other T cell subsets. For example, reduction in gd T cells may be attributed

to increase in the leukemogenic $\alpha\beta$ T cells (**Supplementary figures 2c and 4b-c**). This is also true for the observed reductions in NKT cells which also develop from the T-lineage. Moreover, in our model, MYC-driven T-lymphoma manifests in the CD4 SP, CD8 SP, CD4 CD8 DN, and CD4 CD8 DP compartments (**Figure, right**), making it impossible for us to decipher the roles played by SP T cells in MYC-driven T cell lymphomagenesis.

- (3) The authors state that “NK cell suppression during MYC-driven lymphomagenesis is systemic” through the analysis of NK cells in the BM, blood and spleen that may all contain drastic elevation of lymphoma T cells in MYC^{ON} tumor bearing mice. It would be interesting to determine in other NK cell compartments such as liver and lung that may contain less tumors whether similar alteration in NK cell numbers/development occurs.

The Reviewer raises an important and interesting question regarding the status of NK cells in tissue microenvironments other than the spleen, blood and bone-marrow during MYC-driven T cell leukemogenesis in our T-lymphoma transgenic model. This concern was also raised by Reviewer 3 in minor point 2. To address this concern, we compared by flow cytometry the phenotype of NK cells in liver and lung upon MYC-driven T-lymphoma development in MYC-ON mice, and compared these to age- and strain- matched healthy mice (**Supplementary figures 10 and 11, Manuscript lines 276-288**). Of note, liver and lung of MYC-ON mice exhibit infiltration of T-lymphoblasts (**Supplementary figure 10**), which is accompanied by a concomitant reduction in percentage of total NK cells in comparison to healthy mice (**Supplementary figures 11a-b**). Further analyzing the relative distributions of the NK cells in different stages of maturation, we observed reduction in mNK cell frequencies and increased relative frequency of NKP and iNK (**Supplementary figures 11c-d**) similar to that observed in the spleen (**Figure 3f**). These findings further corroborate that suppression of NK cell maturation occurs in all organs that exhibit MYC^{High} T-lymphoma infiltration.

- (4) The authors demonstrated the dependency of NK suppression on MYC status, by comparing percentages NK cells in MYC OFF mice with residual blasts alongside lymphoma-bearing mice MYC ON with similar blast percentages. addressing whether NK cell suppression is the consequence of lymphoma burden or MYC expression/absence is a key point of this study. To strengthen their result I would suggest the authors to add correlations between NK cell percentages/numbers (or NKp46 MFI) and lymphoma percentages/numbers in MYCOFF and MYCON tumors. We would expect NK cells to be inversely correlated with MYCON but not MYCOFF tumors. With regards to MYCOFF tumors persistence, is there any correlation between NK cell percentages and tumor persistence?

We thank the Reviewer for this comment. As suggested by the Reviewer, we performed correlative analysis between MYC expression levels and numbers of NK cells in MYC-ON and MYC-OFF mice (**Figure below**). We do not observe a statistically significant correlation between MYC levels and NK frequency in either MYC-ON or MYC-OFF groups, when analysed separately (**Figure below**). We would like to kindly point to the Reviewer that a correlation between MYC and NK cells even in the MYC-ON condition is not expected. This is because there is a threshold level of MYC

expression in normal T-lymphocytes (MYC^{Low})

which when surpassed will always lead to MYC^{High} T-lymphoma, and consequently suppress NK cells. MYC

inactivation restores the levels of MYC to those seen in healthy mice, and reverses the suppression of NK cells seen in MYC-ON mice. Hence, MYC

expression level in individual mice of the MYC-ON cohort cannot be employed as an independent predictor of

either disease severity or changes that occur in the immune microenvironment. Furthermore, the absence of a correlation between lymphoblast numbers and NK numbers in the MYC-ON mice suggests that NK cell suppression in MYC-driven lymphomas occurs independently of tumor burden, and not because NK cells are crowded out by excessive T-lymphoblast proliferation (high T-lymphoblast numbers).

Is the tumor persistence affected by NK cell absence, experiments using depleting antibodies (e.g anti-AsGM1 mAbs) could be performed. This is a particularly important point as the authors imply that NK cell alteration upon lymphoma development impact lymphoma development.

We appreciate the Reviewer's concern regarding whether T-lymphoma development is dependent on NK cell absence. Of note, this question was also raised by the Reviewer in point 1. We have carried out a new *in vivo* experiment to study the requirement of NK cells in delaying T-lymphoma development by depleting NK cells in primary 4-week-old SR α -tTA-MYC ON mice using anti NK1.1 antibody prior to overt lymphoma development at 8-12 weeks, and comparing these to IgG2a-treated control mice (**Supplementary figure 22**). We observed that all anti NK1.1-treated

mice succumbed to aggressive lymphomas accompanied by splenomegaly (**Supplementary figure 22b-c**), while control antibody treated mice showed a heterogeneous pattern of disease aggressivity with only 50 percent of the mice developing splenomegaly. Comparison of Disease Free Survival Probabilities indicated slightly prolonged survival in the control group in comparison to the NK-depleted group albeit this was not statistically significant (**Supplementary figure 22a**). We predict that increasing the sample sizes to include more mice per group could lead to statistically significant changes in DFS between the two groups. Of note, we did observe a statistically significant increase in the survival probabilities of mice treated with the control antibody when morbidity was measured as a function of splenomegaly which is indicative of high disease severity (**Supplementary figure 22b**). Hence, we conclude that NK cells are required for blocking MYC-driven lymphomagenesis (**Manuscript lines 453-467**).

- (5) The authors found that mice bearing overt MYC-driven lymphomas have reduced numbers of NK cells because of a developmental blockade at the NK precursor (NKP) stage that prevents the production of immature NKp46+ iNK cells. They compared the absolute counts of NK cell precursors (NKP, CD122+ NKp46-, Figure 3a), and the more differentiated CD122+ NKp46+ iNK (immature NK, Figure 3b) cells in bone marrow from normal (n = 8), EuSR α -tTA MYCON (lymphoma, n = 8), and EuSR α -tTA MYCOFF (regressed lymphoma, n = 8) mice. Given the absence of proper markers assessing NK cell maturation (CD11b, CD27, KLRG1; Huntington ND et al, J Immunol. 2007 Apr 15;178(8):4764-70.; Chiossone L, et al, Blood. 2009 May 28;113(22):5488-96. doi: 10.1182) I would strongly recommend to avoid the term of iNK as the CD112+NKp46+ NK cells, even in the bone marrow, contain true iNK (CD27+CD11b-, Ly49-, NKG2A+) but also mature NKs (CD11b+CD27+ and CD11b+CD27-).

We thank the Reviewer for raising this point and acknowledge that the markers we used to define iNK and mNK were insufficient to correctly identify these populations. This concern was also raised by Reviewer 3 in major point 1. In the revised manuscript, we have redone the entire experiment (**Figure 3, Manuscript lines 223-275**). We observe reduction in the numbers of total NK (lin-CD122+), NKP (lin-CD122+NKp46-CD49b-), iNK (lin-CD122+NKp46+CD49b-) and mNK (lin-CD122+NKp46+CD49b+) in the bone marrow of MYC-ON mice in contrast to normal and MYC-OFF groups (**Figures 3a and 3c**). These findings suggest a block very early in NK cell differentiation at the pre-pro NK stage in the bone marrow which is the primary site of early NK development. Upon analyzing the relative distributions of the NKP, iNK and mNK cells within the lin-CD122+ subset in the bone marrow, we observed a reduction in the relative frequencies of mNK and concomitant increases in the frequencies of NKP and iNK (**Figure 3e**). We observed that this drastic reduction in the mNK fraction in the bone marrow is translated to the periphery (spleen, **Figure 3f**) in MYC-ON mice.

Given the functional differences between iNK and mNK, it is critical to understand whether NK cell maturation stages are differentially affected by MYC. A CD11b CD27 assessment of NK cells in the BM, spleen, liver and lung are also interesting as they have respectively more CD27SP or CD11B SP) of MYCon and MYCoff tumors should be done.

The Reviewer further suggests to also examine other markers of NK maturation including CD27 and CD11b, which was also suggested by Reviewer 3 in major point 2. We examined expression of CD27 and CD11b on mNK cells (Lin-CD122+NKp46+CD49b+) from the spleen, and similar to what has been demonstrated by others (Chiossone *et al.*, Blood, 2009, PMID: 19234143; Hayakawa et al., JI, 2006, PMID:16424180), we identified four distinct functional stages of NK maturation. Within the mNK subset, we observed significantly reduced frequencies of the earliest stage CD27-CD11b- and increased frequencies of terminally differentiated CD27-CD11b+ effector NK cells (**Figures 3h-i**) in MYC-on mice compared to normal and MYC-off mice. Terminally differentiated CD27-CD11b+ NK cells have been previously demonstrated to exhibit reduced functionality in comparison to CD27+CD11b+ NK cells (Hayakawa et al., JI, 2006, PMID:16424180). Hence, our

findings suggest that the few residual NK cells that remain in MYC-on mice may be functionally impaired. Interestingly, we also observed a significantly increased frequency of less-terminally differentiated and active CD27+CD11b+ NK effectors in MYC-OFF T-lymphoma mice (Figures 3h-i), suggesting that MYC inactivation may rescue NK cell function and induce T-lymphoma regression. As suggested by the Reviewer, we also examined expression of CD27 and CD11b on mNK cells in the liver and lung (Supplementary figure 11). Although the group sizes were too small (n = 3 in each group) to make conclusions based on statistics, we did observe a tendency towards decreased frequency of the CD27-CD11b- and increased frequency of CD27-CD11b+ mNK cells in the lungs of MYC-on mice in comparison to healthy mice similar to that observed in the spleen (Supplementary figure 11e).

In addition, although the authors focused mainly on NKp46 the expression of other key NK cell activating (NKG2D, DNAM-1 for example; Martinet L et al Nat Rev Immunol. 2015 Apr;15(4):243-54. doi: 10.1038/nri3799.) or inhibitory receptors (NKG2A, Ly49s) should be performed to determine more extensively the consequence of MYC driven NK cell alteration.

As suggested by the Reviewer we also examined the expression of NKG2D as another key NK cell marker. Identical to that observed for NKp46 (Figures 1, 2), we found a reduction in both frequency of cells expressing NKG2D and reduced MFI in the MYC-on mice compared to Normal and MYC-off mice (Figure left). The data shown in the Figure below further support our conclusion that there is a blockade very early in NK cell differentiation in MYC-driven T-lymphoma.

Again correlation of NK cell maturation stages and tumor burden in MYCON and OFF should be shown to understand the relative importance of MYC expression/tumor burden on NK cell development. In addition functional analysis of NK cells upon MYC tumor development may be decisive to demonstrate MYC driven NK cell dysfunctions.

The Reviewer also suggests determining whether there exists a correlation between NK cell maturation stages and tumor burden in MYC-ON and MYC-OFF mice. As suggested by the Reviewer, we plotted correlation plots between MYC expression levels and the percentages of NKP, iNK and mNK in spleens of MYC-ON and MYC-OFF mice, and observed no significant correlation between any of the pairs (Figure, right).

As explained in our response to Question 4 of the Reviewer, we speculate that there is a threshold level of MYC expression in normal T-lymphocytes (MYC^{Low}) which when surpassed will always lead to T-lymphoma (MYC^{High}), and consequently suppress NK maturation.

The absence of a correlation between MYC and NK cells in the MYC-ON mice suggests that although

individual MYC-driven T-lymphoma mice express varying levels of MYC, these levels are higher than the threshold level observed in healthy mice. Hence, MYC expression in individual mice of the MYC-ON cohort cannot be employed as an independent predictor of either disease severity or changes that occur in the immune microenvironment.

- (6) The authors show that MYC-driven lymphomas evade NK cell-mediated immune surveillance by transcriptionally repressing Type I IFN cytokine axis required for NK cell maturation. However, previous studies including those cited by the authors do not show a critical impact of IFN type I signaling on NK cell frequency and most of them only revealed minor developmental alterations. IFN α R signaling did not affect NK cell number or frequency and led to slightly iNK decrease and mNK increase according to Guan, J. et al (PLOS ONE 9, e111302 2014). Mizutani T et al (Oncoimmunology. 2012 Oct 1;1(7):1027-1037.) showed no significant differences in NKP, NKi and NKm percentages in the bone marrow of Ifn α 1^{-/-} mice. In the spleen they found increased iNK CD27⁺ CD11b⁻ and decreased mNK CD27⁻ CD11b⁺ suggesting that IFN α R deficiency may block NK cell final maturation stage rather than blocking the transition from NKP to NKi. Swann et al (J Immunol. 2007 Jun 15;178(12):7540-9.) did not find striking alteration in the expression of the maturation markers (CD122, CD11b, and Ly49 CI). These studies using IFN α R KO mice questions whether IFN-I alteration upon MYC activation could

be the main mechanism leading to the strong NK cell decrease. To confirm their hypothesis the authors should perform experiments using anti-IFN α R mAbs to determine whether IFN α R blockade prevent NK cell replenishment upon MYC inactivation in their MYCon MYCoff model.

The Reviewer raises an interesting and important question regarding the role of Type I IFN in impacting overall NK cell frequency and the replenishment of NK cells post MYC inactivation. A similar point was raised by Reviewer 3 in major point 4. To interrogate whether Type I IFN signaling is required for restoring the homeostatic frequencies of the NK fractions during MYC inactivation, we treated overt T-lymphoma-bearing MYC-ON mice with IFN α R1 blocking antibodies at the time of MYC inactivation, and examined splenic NK cell maturation after 4 days (**Figures 4h-i and Supplementary figure 14, Manuscript lines 348-362**). We observed no differences in the numbers of total splenic lin-CD122+ NK cells, and the relative frequencies of NKP, iNK and mNK fractions within the lin-CD122+ subset between IFN α R1 blocking antibody-treated and control-treated groups (**Supplementary Figure 14**). However, interestingly, upon examining the frequencies of splenic mNK subsets, we observed a significant decrease in the frequency of CD27+CD11b+ effector cells upon blocking IFN α R1 during MYC inactivation (**Figure 4h-i**). Based on our findings, we conclude that albeit Type I IFN signaling is required for generation of functional NK effectors during late stages of NK maturation, it is not required for early NK maturation and restoration of normal NK numbers upon MYC inactivation.

- (7) In addition in the same study Mizutani showed NK cells maturation defects in *Ifn α 1^{-/-}* mice and not in *Ifn α 1^{f/f} Ncr1-iCre* mice. This suggests that Type I IFN is essential for NK cell maturation but does not require the presence of the Type I IFN receptor on the NK cells themselves, a point that should be at least discussed in the present manuscript since it would suggest that MYC blockade of IFN α signaling affect NK cells indirectly.

We thank the Reviewer for raising this important concern. Of note, this concern was also raised by Reviewer 3 in major point 3. As advised by the Reviewer, we conducted new experiments systematically delineating early (bone marrow) and late (splenic) stages of NK maturation (**Figure 3, Manuscript lines 223-275**), which suggest that development defects seen in NK cells in MYC-ON mice resemble those of NK cells in IFN α R1^{-/-} mice and not those of NK cells in IFN α R1 fl/fl NCR1-Cre mice (Guan et al., Plos One, 2014, PMID: 25333658; Mizutani et al., 2012, PMID: 23170251; Swann et al., 2007, PMID: 17548588):

- First, we observe that MYC-ON mice exhibit reduced numbers of NKP cells during early development in the bone marrow (**Figure 3c**), suggesting a developmental block at the the pre-pro NK stage like that observed in IFN α R1^{-/-} mice by Guan et al. Plos One, 2014, PMID: 25333658.
- Second, we observe reductions in frequencies of mNK (lin-CD122+NKp46+CD49b+) in the spleen (**Figure 3f**) that is characteristic to the IFN α R1^{-/-} mouse (Swann et al., 2007, PMID: 17548588; Mizutani et al., 2012, PMID: 23170251) but not to the *Ifn α 1^{f/f} Ncr1-iCre* mice (Mizutani et al., 2012; PMID: 23170251).
- Third, we observe a disruption at the last four stages of NK differentiation with decreased frequencies of the CD27-CD11b- and increased frequencies of terminally differentiated CD27-CD11b+ splenic mNK cells in MYC-ON mice as compared to normal and MYC-OFF groups (**Figure 3h-i**). Of note, Mizutani et al., Oncoimmunology, 2012, also demonstrate that IFN α R1^{-/-} mice have significant disruptions in the last 4 stages of terminal NK differentiation marked by changes in CD27 and CD11b expression, and that these changes do not occur in IFN α R1 fl/fl NCR-Cre mice.

Since the changes we observe in NK maturation in MYC-ON mice resemble those of IFN α R1-/- mice and not those of IFN α R1 fl/fl NCR1-Cre mice, we conclude (as the Reviewer correctly notes) that IFN α R1-dependent changes in NK maturation in MYC-ON mice possibly do not require the presence of IFN α R1 on the NK cells themselves. Like the Reviewer points out, our results (Figure 3) rather suggest that the suppression of Type I IFN signaling in MYC-ON mice may indirectly affect NK cell maturation by acting on other immune subsets that in turn control NK maturation. Of note, we observe a reduction in IL15 and IL15R in MYC-ON mice (Figure below) suggesting that loss of Type IFN secretion by the MYC-transformed lymphocytes may be affecting DC maturation (Montoya *et al.*, Blood, 2012, PMID: 11964292), and consequently reducing the levels of IL15 (Mattei *et al.*, JI, 2001, PMID: 11466332), a cytokine essential for NK cell maturation and function (Carson *et al.* J Exp Med, 1994, PMID: 7523571).

Consistent with previous studies showing an increase in DCs numbers and a block in DC maturation in MYC-driven B-lymphomas (Rehm *et al.*, Nat Commun, 2014, PMID: 25266931), we do show that DC numbers are significantly increased in MYC-ON mice (Supplementary figure 4e). Respectfully, we believe that formally studying the phenotype and function of DCs and their effects on NK maturation in our inducible MYC-driven T-lymphoma model would constitute an entirely separate study. Hence, we have discussed these possibilities in the manuscript (Manuscript lines 541-552), and will interrogate how DCs affect NK maturation in MYC-driven lymphomas in future studies from our group.

- (8) The authors show that malignant lymphocytes overexpressing oncogenic MYC show suppressed production of Type I IFNs (Figures 5e-f, Supplementary Figures 10, 15). they conclude that silenced type I signaling by lymphoma cells leads to NK cells maturation blockade. Although data showing that MYC can directly affect type I IFN are convincing it is difficult to understand why tumor cell derived type I IFN is required for NK cell maturation since NK cell maturation occurs in the absence of lymphoma in naïve mice. The authors should investigate whether lymphoma expressing myc can also affect type I IFN production by cells from the microenvironment (DC for example). It is reasonable to speculate that type I IFN silencing by MYC could limit tumor immunogenicity leading to poor NK cell priming and killing but it is difficult to understand why tumor derived IFN I is required for NK cell development especially if type I IFN are normally produced by other cells from the microenvironment. The authors should at least discuss alternative hypothesis. For instance IL-15 was previously shown to be critical for NK cell development and priming (Huntington ND, J Exp Med. 2009 Jan 16;206(1):25-34.) and IL-15 was shown to be induced by type I IFN on DCs (Ganal SC, Immunity. 2012 Jul 27;37(1):171-86. doi: 10.1016/j.immuni.2012.05.020). IL-15 shortage upon lymphoma development should be envisaged.

The Reviewer raises a thoughtful and important question. We agree with the Reviewer that although our data strongly suggest the loss of Type I IFN production by MYC-transformed B- (**Figure 5f and supplementary figure 17**) and T- cells (**Supplementary figure 19c**), we do not exclude the possibility that MYC-driven leukemogenesis disrupts the homeostasis of key Type I IFN producing immune subsets including Dendritic Cells (DCs, Montoya *et al.*, Blood, 2012, PMID: 11964292). As the Reviewer correctly points out IL15 production by DCs (Mattei *et al.*, JI, 2001, PMID: 11466332) could be impaired in MYC-ON mice leading to the defects in NK cell maturation (Carson *et al.* J Exp Med, 1994, PMID: 7523571) we observe (**Figure 3**). Of note, we observe upon bulk RNA sequencing of splenic samples from normal, MYC-ON, and MYC-OFF mice that IL15 and IL15R are significantly reduced in MYC-ON mice (**Figure below**). These findings suggest that sources of lost Type I IFN may be other immune subsets in addition to the lymphoblasts themselves. We have now discussed this possibility in the manuscript (**Manuscript lines 541-552**).

Reviewer #3:

Swaminathan et al., report that MYC drive T cell lymphomas drive developmental defects in splenic B and NK cells. Inducible repression of MYC rescues NK and B cell development. The authors propose a mechanism by which MYC-driven lymphomas decrease type I interferon production to block NK cell development and subsequent tumor surveillance. Cytokine treatment with IFN- α is sufficient to restore NK cell numbers and increase survival in MYC ON mice. However, there are quite a few key issues with the proposed hypothesis, with additional detail needed to resolve these issues:

We thank the Reviewer for taking the time to review our manuscript and for making important suggestions to markedly improve our study. The Reviewer's suggestions have enabled us to gain a better understanding of how the MYC oncogene causally regulates NK surveillance in lymphoid malignancies. Below, we describe the new experiments and analyses we conducted to address the Reviewer's specific concerns. Please note that all **New Figures** in the revised manuscript are highlighted in blue and grey throughout the rebuttal. Please also note that Figures not included in the final manuscript are shown in response to the Reviewers specific concerns

Major Issues

- (1) The authors do not correctly identify iNK, NKP, and mature NK cell populations in the spleen and bone marrow. NKP = lin⁻ (CD19, CD3, TCR α) CD122+NKp46-NK1.1⁻, iNK = lin⁻CD122+NKp46+NK1.1+CD49b⁻, mNK = lin⁻CD122+NKp46+NK1.1+CD49b⁺. These errors are apparent in Fig 3H with a drastic increase in the proposed "NKP" population, which would likely be gated out using lineage markers. The data in figure 3 are insufficient to correctly identify these populations and need to be redone.

We thank the Reviewer for raising this point and acknowledge that identification of NKP, iNK and mNK in bone marrow and spleen was done incorrectly. In the revised manuscript, we have redone the entire experiment (**Figure 3, Manuscript lines 223-247**) using the gating strategy kindly provided by the Reviewer (**Supplementary figure 25b**). We observe reduction in the numbers of total NK (lin-CD122⁺), NKP (lin-CD122+NKp46-CD49b⁻), iNK (lin-CD122+NKp46+CD49b⁻) and mNK (lin-CD122+NKp46+CD49b⁺) in the bone marrow of MYC-ON mice in contrast to normal and MYC-OFF groups (**Figure 3a and 3c**). These findings suggest a block very early in NK cell differentiation at the pre-pro NK stage in the bone marrow which is the primary site of early NK development. Upon analyzing the relative distributions of the NKP, iNK and mNK cells within the lin-CD122⁺ NK subset in the bone marrow, we observed a reduction in the relative frequencies of mNK and concomitant increases in the frequencies of NKP and iNK (**Figure 3e**). We observed that this drastic reduction in the mNK fraction in the bone marrow is translated to the periphery (spleen, **Figure 3f**) in MYC-ON mice.

- (2) The authors claim that NK cell development is perturbed and need to strengthen these data by including other markers commonly used to measure NK cell maturation in addition to NKp46 (CD11b, CD27, KLRG1).

We thank the Reviewer for this suggestion which was also made by Reviewer 2 in point 5. To address the Reviewer's concern, we have included additional markers of NK maturation to strengthen our findings (**Figure 3, Manuscript lines 248-275**). We examined expression of CD27 and CD11b on mNK cells (Lin-CD122+NKp46+CD49b⁺) from the spleen, and like that demonstrated by others (Chiossone *et al.*, Blood, 2009, PMID: 19234143; Hayakawa *et al.*, JI, 2006, PMID:16424180), we identified four distinct functional stages of NK maturation. Within the mNK subset, we observed significantly reduced frequencies of the earliest stage CD27-CD11b⁻ and increased frequencies of terminally differentiated CD27-CD11b⁺ effector NK cells (**Figure 3h**) in

MYC-on mice in comparison with Normal and MYC-off mice. Terminally differentiated CD27-CD11b+ NK cells have been previously demonstrated to exhibit reduced effector function in comparison to CD27+CD11b+ NK cells (Hayakawa et al., JI, 2006, PMID:16424180). Hence, our findings suggest that the few residual NK cells that remain in MYC-on mice may be functionally impaired. Interestingly, we also observed a significantly increased frequency of less-terminally differentiated and active CD27+CD11b+ NK effectors in MYC-off T-lymphoma mice (**Figures 3h-i**), suggesting that MYC inactivation may rescue NK cell function and induce T-lymphoma regression.

As suggested by the Reviewer we also compared the expression of KLRG1 on the mature splenic NK cells in Normal, MYC-on and MYC-off mice and found no statistical significant differences in the frequencies of KLRG1+ NK cells across the groups. These data are presented for the Reviewer in the **Figure on the right**.

The authors incorrectly claim that previous studies show a direct role for type I interferon in NK cell development. These studies show that NK cell frequencies and numbers are largely similar between WT and IFN α R-deficient mice, and that there is no cell-intrinsic role of type I interferons in NK cell development (Madera et al., 2016 JEM; Guan et al., 2014 Plos One; Baranek et al., 2012 Cell Host Microbe) These data contrast to the developmental defects observed in the MYC ON mice. Thus, the possibility remains that in vivo IFN- α treatment has cell-extrinsic effects on NK cell development and or proliferation in the MYC ON state, possibly by driving increased IL-15 production in vivo. Because both IL-15 and IL-15ra expression are increased in the spleens of WT and MYC OFF mice, and DC homeostasis is not perturbed in the MYC ON state, the author's need to formally address or disprove the possibility that decreased systemic levels of IL-15 can explain their defects in NK development.

We thank the Reviewer for raising this important concern. Of note, this concern was also raised by Reviewer 2 in point 7. As advised by the Reviewer, we conducted new experiments systematically delineating early (bone marrow) and late (spleen) stages of NK maturation (**Figures 3, Manuscript lines 223-275**), which suggest that development defects seen in NK cells in MYC-ON mice resemble those of NK cells in IFN α R1^{-/-} mice and not those of NK cells in IFN α R1 fl/fl NCR1-Cre mice (Guan et al., Plos One, 2014, PMID: 25333658; Mizutani et al., 2012, PMID: 23170251; Swann et al., 2007, PMID: 17548588):

- First, we observe that MYC-ON mice exhibit reduced numbers of NKP cells during early development in the bone marrow (**Figure 3c**), suggesting a developmental block at the the pre-pro NK stage like that observed in IFN α R1^{-/-} mice by Guan et al. Plos One, 2014, PMID: 25333658.
- Second, we observe reductions in frequencies of mNK (lin-CD122+NKp46+CD49b+) in the spleen (**Figure 3f**) that is characteristic to the IFN α R1^{-/-} mouse (Swann et al., 2007, PMID: 17548588; Mizutani et al., 2012, PMID: 23170251) but not to the Ifn α 1f/f Ncr1-iCre mice (Mizutani et al., 2012, PMID: 23170251).
- Third, we observe a disruption at the last four stages of NK differentiation with decreased frequencies of the CD27-CD11b- and increased frequencies of terminally differentiated CD27-CD11b+ splenic mNK cells in MYC-ON mice as compared to normal and MYC-OFF

groups (**Figure 3h**). Of note, Mizutani et al., *Oncoimmunology*, 2012, also demonstrate that IFN α R1^{-/-} mice but not the IFN α R1 fl/fl NCR-Cre mice exhibit significant disruptions in CD27-CD11b⁻ and CD27-CD11b⁺ splenic mNK cells.

Since the changes we observe in NK maturation in MYC-ON mice resemble those of IFN α R1^{-/-} mice and not IFN α R1 fl/fl NCR1-Cre mice, we conclude (as the Reviewer correctly notes) that IFN α R1-dependent changes in NK maturation in MYC-ON mice possibly do not require the presence of IFN α R1 on the NK cells themselves. Like the Reviewer points out, our results (**Figure 3**) rather suggest that the suppression of Type I IFN signaling in MYC-ON mice may indirectly affect NK cell maturation by acting on other immune subsets that in turn control NK maturation. Of note, we observe a reduction in IL15 and IL15R in MYC-ON mice (**Figure below**) suggesting that loss of Type IFN secretion by the MYC-transformed lymphocytes may be affecting DC maturation (Montoya *et al.*, *Blood*, 2012, PMID: 11964292), and as a consequence reducing the levels of IL15

(Mattei et al., *J Immunol*, 2001, PMID: 11466332), a cytokine essential for NK cell maturation (Carson et al. *J Exp Med*, 1994, PMID: 7523571).

Consistent with previous studies showing an increase in DCs numbers and a block in DC maturation in MYC-driven B-lymphomas (Rehm *et al.*, *Nat Commun*, 2014, PMID: 25266931), we do show that DC numbers are significantly increased in MYC-ON mice (**Supplementary figure 4e**). Respectfully, we believe that formally studying the phenotype and function of DCs and their effects on NK maturation in our inducible MYC-driven T-lymphoma model would constitute an entirely separate study. Hence, we have discussed these possibilities in the manuscript (**Manuscript lines 541-552**), and will interrogate how DCs affect NK maturation in MYC-driven lymphomas in future studies from our group.

(3) The authors claim that MYC driven T cell lymphomas suppress type I interferon production to perturb NK cell development. The authors need to demonstrate that deletion of type I interferons, specifically in MYC OFF lymphomas, is required for the observed rescue in NK cell development.

We thank the Reviewer for this thoughtful suggestion to conduct an important functional experiment to strengthen our study. Of note, this concern was also raised by Reviewer 2 in point 6. To interrogate whether Type I IFN signaling is required for restoring the homeostatic frequencies of the NK fractions during MYC inactivation, we treated overt T-lymphoma-bearing MYC-ON mice with IFN α R1 blocking antibodies at the time of MYC inactivation, and examined splenic NK cell maturation after 4 days (**Figures 4h-i and Supplementary figure 14, Manuscript lines 348-362**). We observed no differences in the numbers of total splenic lin-CD122⁺ NK cells, and the relative frequencies of NKP, iNK and mNK fractions within the lin-CD122⁺ subset between IFN α R1 blocking antibody-treated and control-treated groups (**Supplementary figure 14**). However, interestingly, upon examining the frequencies of splenic mNK subsets, we observed a significant decrease in the frequency of CD27⁺CD11b⁺ effector cells upon blocking IFN α R1 during MYC inactivation (**Figures 4h-i**). Based on our findings, we conclude that albeit Type I IFN signaling is

required for generation of functional NK effectors during late stages of NK maturation, it is not required for early NK maturation and restoration of normal NK numbers upon MYC inactivation.

Minor

(1) The authors need to demonstrate that there is no effect of doxycycline treatment on NK cell development (in WT mice). This is important because previous studies have shown that acute antibiotic treatment can impact NK cell homeostasis (Kamimura et al., 2015 Cell Reports).

We thank the Reviewer for raising this concern. To rule out off-target effects of doxycycline treatment on NK cell development, we treated healthy control mice (age- and strain-matched to SR α -tTA-Tet-O-MYC mice) with doxycycline for 4 days and examined NK cell development and maturation in bone-marrow and spleen (**Supplementary figure 9, Manuscript lines 264-275**). We did not observe statistical significant differences in NKP, iNK and mNK cells in bone-marrow or spleen between doxycycline treated and control mice (**Supplementary figures 9a-h**). We also examined expression of CD27 and CD11b on mNK cells in the spleen and did not observe any statistical significant differences between doxycycline treated and control mice (**Supplementary figure 9i-j**). We therefore conclude that doxycycline treatment is not responsible for the observed changes in NK cell development and maturation in MYC-OFF T-lymphoma mice.

(2) The authors should look in the liver and lung of MYC ON and MYC OFF mice to determine if the decrease in NK cells is not confined to discrete tissue microenvironments (spleen and bone marrow).

The Reviewer raises an important and interesting question regarding the status of NK cells in tissue microenvironments other than the spleen, blood and bone-marrow during MYC-driven T cell leukemogenesis in our T-lymphoma transgenic model. Reviewer 2 also raises this concern in point 3. To address this concern, we compared by flow cytometry the phenotype of NK cells in liver and lung upon MYC-driven T-lymphoma development in MYC-ON mice, and compared these to age-matched healthy mice (**Manuscript lines 276-288**). Of note, liver and lung of MYC-ON mice exhibit infiltration of T-lymphoblasts (**Supplementary Figure 10**), which is accompanied by a concomitant reduction in percentage of total NK cells in comparison to healthy mice (**Supplementary Figures 11a-b**). Further analyzing the relative distributions of the NK cells in different stages of maturation, we observed reduction in mNK cell frequencies and increased relative frequency of NKP and iNK (**Supplementary Figures 11c-d**) similar to that observed in the spleen (**Figure 3f**). These findings further corroborate that suppression of NK cell maturation occurs in all organs that have T-lymphoma infiltration.

REVIEWERS' COMMENTS:

Reviewer #1 (Remarks to the Author):

Concerns have been addressed and manuscript is ready for publication.

Reviewer #2 (Remarks to the Author):

The study from Swaminathan et al showing that myc oncogenic pathway in tumors restrict NK cell anti-tumor immune response by repressing type I IFN signaling in the tumor microenvironment is of real interest for a broad readership. The authors correctly addressed most of the questions and considerably improved their manuscript during the reviewing process. I only have one question below that may further add to the manuscript by the authors.

The authors confirmed that oncogenic MYC blocks both early and late stages of NK cell maturation during lymphomagenesis. They brought additional data showing that MYC suppresses NK cells in two ways: -First, overexpression of MYC in T-lymphoblasts causes reduction in numbers of all linCD122+ NK cells in the bone marrow.

-Second, MYC overexpression in T-lymphoblasts of MYC-ON mice perturbs the homeostatic distributions of residual mature splenic NK effector subsets.

The profile of the dot plots in figure 3i and 4i, 9i are very odd. The percentage of CD11b-CD27- is abnormally high (40%) even in the non-tumor bearing mice. It is widely accepted that only few NK cell are CD11b-CD27- whereas most of the NK are CD11b+CD27+ or CD11b+CD27- (cf figure 1a from Chiossone L, et al, Blood. 2009 May 28;113(22):5488- 96. doi: 10.1182). This may reveal inappropriate gating strategy used to detect NK cell maturation or alternatively a poor CD11b staining. The authors should try to correct these findings.

Reviewer #3 (Remarks to the Author):

All major concerns have been addressed

Reviewer#2

The study from Swaminathan et al showing that myc oncogenic pathway in tumors restrict NK cell anti-tumor immune response by repressing type I IFN signaling in the tumor microenvironment is of real interest for a broad readership. The authors correctly addressed most of the questions and considerably improved their manuscript during the reviewing process. I only have one question below that may further add to the manuscript by the authors.

The authors confirmed that oncogenic MYC blocks both early and late stages of NK cell maturation during lymphomagenesis. They brought additional data showing that MYC suppresses NK cells in two ways: -First, overexpression of MYC in T-lymphoblasts causes reduction in numbers of all linCD122+ NK cells in the bone marrow. -Second, MYC overexpression in T-lymphoblasts of MYC-ON mice perturbs the homeostatic distributions of residual mature splenic NK effector subsets.

The profile of the dot plots in figure 3i and 4i, 9i are very odd. The percentage of CD11b-CD27- is abnormally high (40%) even in the non-tumor bearing mice. It is widely accepted that only few NK cell are CD11b-CD27- whereas most of the NK are CD11b+CD27+ or CD11b+CD27- (cf figure 1a from Chiossone L, et al, Blood. 2009 May 28;113(22):5488- 96. doi: 10.1182). This may reveal inappropriate gating strategy used to detect NK cell maturation or alternatively a poor CD11b staining. The authors should try to correct these findings.

We appreciate the Reviewer's valid concern regarding the expression pattern of CD11b+ NK cells in wild type mice in our studies compared to that demonstrated by Chiossone et al. We had identical concerns while conducting our study because the expression values of both CD27 and CD11b on mature NK cells in our normal mice do not match that shown in the literature (Chiossone et al., 2009).

We speculated that these discrepancies may be due to strain-mediated differences in NK cell markers between C57BL/6J mice (used by Chiossone et al.) and FVB/N mice which is the background for MYC-driven T-lymphoma mice (used in our current study). To test this hypothesis, we conducted a side-by-side comparison of CD11b and CD27 expression in NK cells of FVB/N and C57BL/6J wt mice. (See ppt file **Response to Reviewer 2: Figure 1**). Indeed, we observed significant differences in the expression patterns of CD27 and CD11b on the mature NK cells between FVB/N and C57BL/6J mice with increased frequencies of the CD27-CD11b- and the CD27+CD11b- and decreased frequencies of the CD27+CD11b+ and CD27-CD11b+ populations in the FVB/N mice compared to the C57BL/6J mice. Of note, as the Reviewer correctly points out, FVB/N mice had a high percentage of the CD27-CD11b- NK cells whereas only few NK cells in the C57BL/6J mice are CD27-CD11b-. We believe the observed strain-specific differences in the expression patterns of CD27 and CD11b on the mature NK cells between FVB/N and C57BL/6J mice (See ppt file **Response to Reviewer 2: Figure 1**) explain our findings.

We would also like to kindly point out to the Reviewer that our staining is tightly controlled using FMO (fluorescence minus one) as shown in our gating strategy (See ppt file **Response to Reviewer 2: Figure 2c; Manuscript Supplementary Figure 25c**).

Based on these findings, we conclude that the differences between our observations and those of Chiossone et al. reflect differences in effector NK subset frequencies between strains of mice.

Response to Reviewer 2: Figure 1

Strain-specific expression pattern of CD27 and CD11b on mature NK cells.

(a-b) Quantification of percentages (a) and representative flow plots (b) of mNK cells expressing CD27 and CD11b in spleen in FVB/N wt (n = 3) and C57BL/6J wt (n = 5) mice. Populations are gated on Lin-CD122+NKp46+CD49b+ mature NK (mNK) intact live singlets. P-values have been calculated using the Mann-Whitney test (*p < 0.05)

Supplementary Figure 25: Gating strategy for CyTOF and flow cytometry and FMO controls